



# Temporal and Spatial Influences of Environmental Factors on the Distribution of Mesopelagic organism in the North Atlantic Ocean

Jie Yang[1,2,*], Jian Hui Li[1,2,*], and Ge Chen[1,2]

[1]Department of Marine Technology, Ocean University of China, Qingdao, China
[2]Laoshan Laboratory, Qingdao, China
[*]These authors contributed equally to this work.

**Correspondence:** Ge Chen (gechen@ouc.edu.cn)

**Abstract.** Mesopelagic organisms play a critical role in marine ecosystems and the global carbon cycle, acting as key intermediaries between trophic levels through diel (DVM) and seasonal vertical migrations (SVM). However, the seasonal vertical migration patterns of these organisms, and the environmental drivers influencing them, remain insufficiently understood. Here, we analyzed 83,603 backscattering coefficient ($b_{bp}$) profiles obtained from 720 BGC-Argo floats deployed in the North Atlantic

Ocean from 2010 to 2021. This extensive dataset enabled the identification of spiking layer signals, allowing us to investigate the diurnal and seasonal vertical distributions of mesopelagic organisms, as indicated by these $b_{bp}$ spikes. Additionally, we examined the horizontal heterogeneity in these distributions and their correlations with key environmental variables. Our findings reveal distinct diurnal migrations, characterized by multilayered aggregations predominantly in the mid-ocean during daylight, with prominent signals at depths around 150 m, 330 m, 650 m, and 780 m. At night, a strong scattering layer forms in the upper

ocean, with signals concentrated at depths shallower than 350 m, particularly in the top 100 m. Seasonal analyses shows that in spring and winter, the average $b_{bp}$ spike intensity is lower in the upper ocean than in the mid-ocean, although the frequency of $b_{bp}$ spikes is higher in the upper ocean. In contrast, summer and autumn—especially summer—exhibit both higher mean $b_{bp}$ spike intensity and frequency near the surface. Spatially, mesopelagic organisms migrate deeper in the northeast and remain shallower in the southwest, correlating with higher temperatures and shallower distributions. Random forest analysis identified

temperature as the most influential environmental factor affecting the distribution of mesopelagic organisms year-round, with the temperature gradient being particularly critical. Other critical factors include seawater salinity, dissolved oxygen, surface chlorophyll concentration, and latitude, with relative importance of 29.44%, 15.49%, 14.85%, 13.46%, and 12.35%, respectively. This study enhances our understanding of the mechanisms driving carbon transfer to the deep ocean and the energy and material cycles within marine ecosystems, providing a basis for future fisheries management in mesopelagic environments.

## 1 Introduction

The mesopelagic organisms, comprising species such as zooplankton, shrimp, squid, fish, and jellyfish, are estimated to harbor around one billion tonnes of biomass, representing a significant fraction of global fish biomass(Irigoien et al., 2014). This biome is a crucial component of marine ecosystems, serving as a vital link between primary producers and higher trophic levels. It plays a fundamental role in the ocean's energy transfer and nutrient cycling(Klevjer et al., 2016; Kruse et al., 2010).





A prominent behavioral adaptation in the mesopelagic zone is diel vertical migration (DVM), wherein organisms undertake extensive vertical movements to optimize survival and foraging efficiency. During daylight, they inhabit depths of several hundred meters in the mesopelagic zone to minimize predation risk, while at night, they migrate to the epipelagic zone to exploit food resources. This migration is recognized as one of the largest-scale migrations on Earth(Kapelonis et al., 2023; Hays, 2003; Petrusevich et al., 2020). Additionally, mesopelagic organisms undergo seasonal vertical migration (SVM), adjusting

their vertical distribution in response to environmental fluctuations. Both DVM and SVM drive the active export of organic and inorganic materials—through excretion, defecation, respiration, and mortality—into deeper ocean layers(Lourenço and Jany, 2021). These processes are not only ecologically important but also play a significant role in biogeochemical cycling, carbon sequestration, and mitigating global climate change(Govindarajan et al., 2023; Hazen, 2022; Gjoesaeter et al., 1980; Hays, 2003; Ramirez-Llodra et al., 2010; Robinson et al., 2010; Bailey, 2021).

Traditional approaches to study mesopelagic organisms have largely relied on trawl and acoustic sampling methods. Although trawl sampling is more frequently used, it suffers from limitations in spatial and temporal resolution, as well as biases related to evasion and selectivity, which impede accurate estimates of migration timing, rate, and extent(Sutton, 2013; Underwood et al., 2020; Luo et al., 2000). Acoustic sampling offers greater precision but is restricted by the spatial and temporal coverage due to platform constraints, such as vessels. The resolution of acoustic sensors often fails to detect small, dispersed,

and weakly scattering species at depth, and the high costs associated with traditional ADCP-based methods further limit extensive in situ observations(Haëntjens et al., 2020; Chai et al., 2020; Underwood et al., 2020; Nakao et al., 2021). Recently, the increased deployment of BGC-Argo floats has enhanced accuracy and broadened the scope of applications in various studies, including inter-annual analyses of phytoplankton communities, quantification of carbon export from the ocean interior, and observations of mesocosm flux decay due to fragmentation processes(Rembauville et al., 2017; Xing et al., 2020; Wang

and Fennel, 2022; Galí et al., 2022; Briggs et al., 2020; Boyd et al., 2019). Bio-optical sensors mounted on these floats have proven effective in detecting a range of bio-optical properties, renderiimportanceerful tools for large-scale spatial detection of mesopelagic organisms(Claustre et al., 2019; Haëntjens et al., 2020).

Recent advancements have demonstrated that backscattering coefficient ($b_{bp}$) spike signals from BGC-Argo floats can effectively infer mesopelagic biological information, showing high concordance with acoustic trawl observations(Haëntjens et al.,

2020). These $b_{bp}$ spike signals are primarily generated by larger particles, which are closely associated with biological aggregations(Briggs et al., 2011). Notably, the extensive diatom blooms in the North Atlantic each spring result in substantial pulses of particulate matter, comprising fresh phytoplankton aggregates that rapidly descend to the seafloor(Lampitt, 1985; Honjo and Manganini, 1993). Occasionally, large peaks in optical profiles are interpreted as aggregates or zooplankton(Bishop et al., 1999; Gardner et al., 2000; Bishop and Wood, 2008). The $b_{bp}$ signal captures the entire particle assemblage, including zooplankton,

detritus, bacteria, and mineral particles. Significant increases in $b_{bp}$ are observed when small zooplankton dominate the mixed layer community(Rembauville et al., 2017; Petit, 2023). Furthermore, satellite-based lidar inversion of $b_{bp}$ signals has shown that zooplankton activity leads to pronounced $b_{bp}$ spikes, particularly at night, with these spikes most evident in the surface ocean layers, revealing the global distribution characteristics and evolutionary patterns of diel vertical migration(Behrenfeld





et al., 2019). Collectively, these studies indicate that $b_{bp}$ spike signals not only reflect the presence of large particulate matter and tiny zooplankton but also capture the diel vertical migration of zooplankton.

However, substantial challenges remain in the large-scale detection of mesopelagic organisms, resulting in significant uncertainties in biomass estimates, which range widely from billions to hundreds of tonnes(Gjoesaeter et al., 1980; Sarant, 2014). Moreover, the patterns of DVM and SVM, their adaptive mechanisms, and the multifactorial influences on these behaviors remain poorly understood(Bandara et al., 2021).

In this study, we extracted $b_{bp}$ spike signals from BGC-Argo floats in the mid- and high-latitude regions of the North Atlantic Ocean to statistically infer the aggregation patterns of mesopelagic organisms. We then investigated diurnal and seasonal vertical migration patterns, analyzed the horizontal spatial distribution characteristics, and evaluated the environmental drivers influencing vertical distribution using Random Forest modeling.

## 2 Material and methods

### 2.1 Study area

The North Atlantic plays a critical role in global carbon sequestration, where pelagic fish are key to marine ecosystem dynamics and represent a largely untapped resource for fisheries(Gruber et al., 2002). The high-latitude overturning circulation, combined with the subduction processes of the Subtropical Circulation (STC), drives the transport of dissolved organic carbon (DOC) to the deep ocean, functioning as a core mechanism of the biological pump and contributing significantly to the global carbon cycle(Hansell et al., 2002). Recent studies have reported notable shifts in species composition and dominance within fish assemblages in the subarctic Northwest Atlantic, especially around the shallow Reykjanes Ridge(Sutton et al., 2008). These changes highlight the potential for profound impacts on marine ecosystem management and the sustainable use of fishery resources. Thus, the North Atlantic is central to addressing global climate change, preserving biodiversity, and guiding the sustainable use of marine resources. This study examines the North Atlantic Ocean, spanning latitudes from 35° to 75°N and longitudes from 0 to 70°W. Figure 1 illustrates the spatial distribution of the backscattering coefficient ($b_{bp}$) profiles collected by BGC-Argo floats across the study area, based on 1°×1°grid statistics.





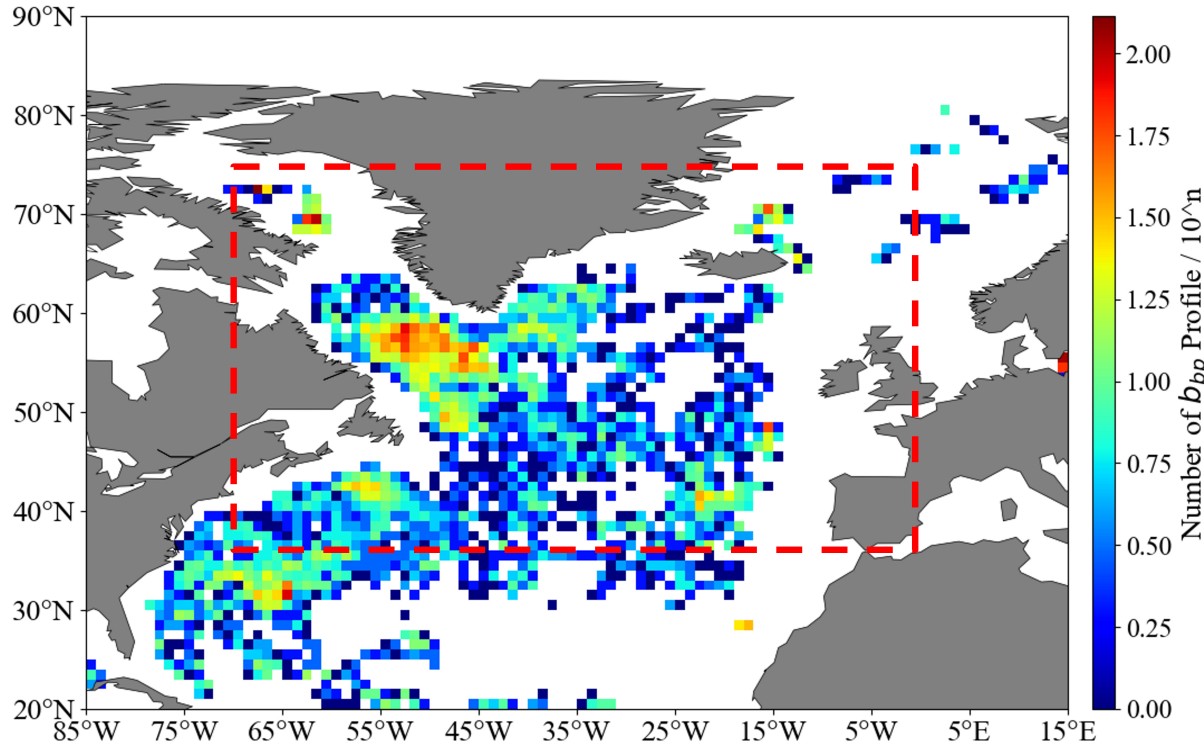

**Figure 1.** Total BGC-Argo Profile Count with $b_{bp}$ Layer at 1°×1°Grid Resolution; Study Area Highlighted in Red.dotted frame

## 2.2 Data

The dataset used in this study includes BGC-Argo profiles and remote sensing data collected from 2010 to 2021. The BGC-Argo dataset consists of profiles from 720 floats, capturing key parameters such as Chlorophyll-a, dissolved oxygen, salinity,

temperature, and particle backscattering at 700 nanometers ($b_{bp}$700). A total of 83,603 valid profiles were selected based on the detection of $b_{bp}$700 spikes. Remote sensing data comprises sea surface temperature (SST) and chlorophyll-a (Chl). SST data, obtained from the National Oceanic and Atmospheric Administration (NOAA), consists of optimally interpolated fields with a spatial resolution of 0.25°, derived from a fusion of Advanced Very High-Resolution Radiometer (AVHRR) observations from multiple platforms, providing high accuracy and broad spatial coverage. The Chl data, sourced from GlobColour, is a Level 3

ocean color product with a 0.25° resolution, combining outputs from multiple chlorophyll sensors to ensure data continuity, enhance spatial and temporal coverage, and reduce noise.





### 2.3  Methods

#### 2.3.1  BGC-Argo spike layer observations method

To investigate the aggregation patterns of mesopelagic organisms, we utilized a previously established spike layer detection
algorithm(Haëntjens et al., 2020)to extract $b_{bp}$ spike signals. The extraction process involved several key steps: Initially, we
screened $b_{bp}$ profiles, selecting those with more than 30 sampling points and a maximum depth greater than 50 meters. The
raw $b_{bp}$ signal of each profile was smoothed using a 15-point Hampel filter, establishing a baseline signal. Next, we computed
the difference between the original $b_{bp}$ signal and the baseline. Signals exhibiting differences exceeding twice the smad were
identified as spike signals.( see Eq. (1) for smad calculation, where smad represents the minimum threshold of the profile and
$b_{bp}$(n) represents all $b_{bp}$ signals in the profile). These detected spike signals were subsequently clustered hierarchically using a
depth parameter of 50 meters, and the results were categorized based on distinct features. Spike signals with identical features
across two profiles were aggregated into a spike layer. For each layer, we quantified the intensity, depth, and spike count, which
were then recorded for further analysis. The spike layer extraction workflow is illustrated in Figure 2.

$$smad = -\frac{1}{\sqrt{2} \cdot erfcinv(\frac{3}{2})} \cdot median(\mid b_{bp}(i) - median(b_{bp}(n)) \mid) \tag{1}$$

where $smad$ represents the minimum threshold of the profile, $b_{bp}(i)$ represents the median value of all signals in the profile,
and $b_{bp}(n)$ represents all $b_{bp}$ values throughout the profile.

We first removed outliers from the environmental profiles and interpolated missing data points. Subsequently, a 25-point
median filter followed by a mean filter was applied to the environmental data to minimize the influence of outliers and missing
values on the analysis accuracy. After preprocessing, we calculated the temperature gradients for each profile, along with the
mean dissolved oxygen and salinity values over depth intervals of 0–200 m, 200–500 m, and 500–800 m. Given the higher
variability in chlorophyll and temperature in the upper and middle layers, we averaged these parameters over depth ranges
of 0–50 m, 50–200 m, and 200–500 m. These averaged values served as environmental inputs for the Random Forest model.
Additionally, sea surface chlorophyll and sea surface temperature data, with a spatial resolution of 0.25 degrees and a temporal
resolution of one day, were integrated with the BGC-Argo profile data to enhance the analysis.



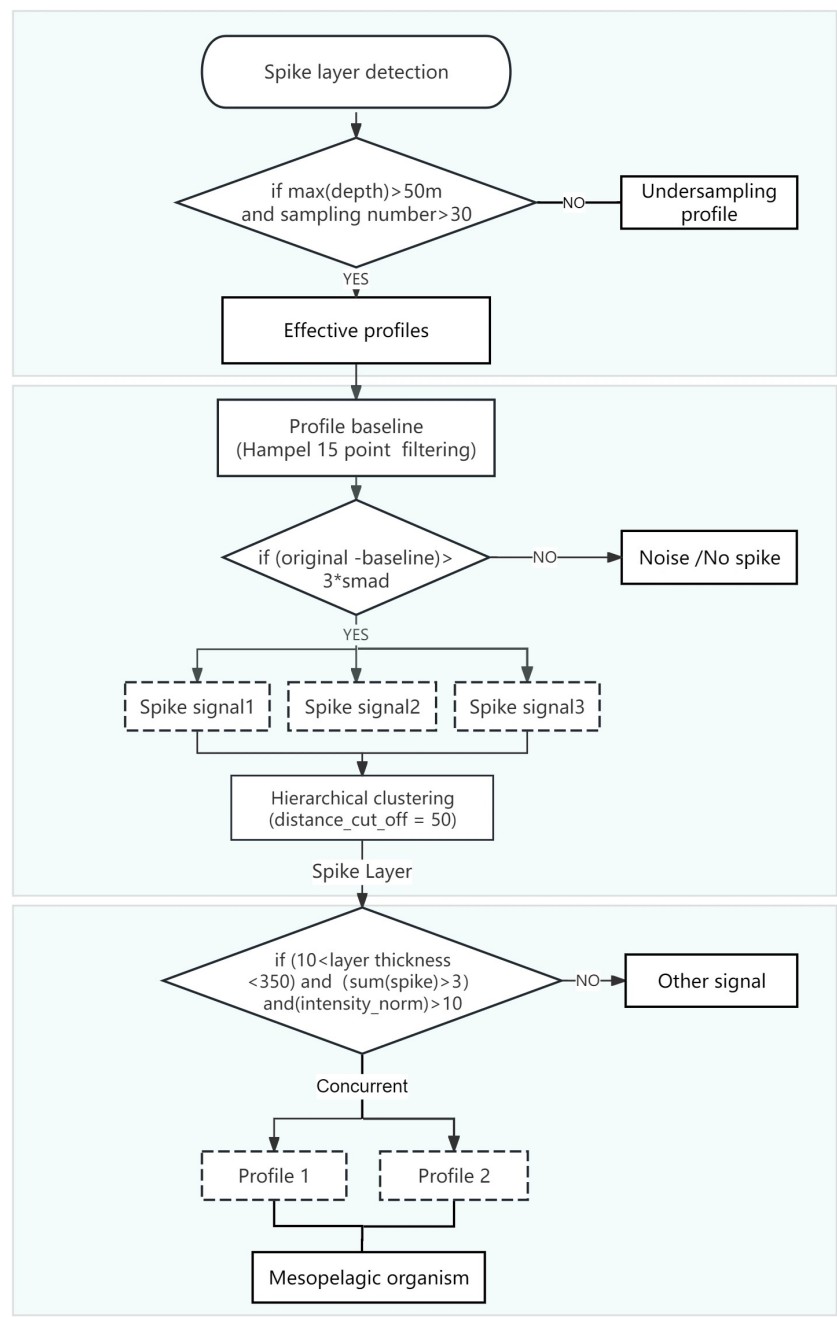

**Figure 2.** Extraction process of spike layer signal.



### 2.3.2   Statistical Method

To elucidate the diurnal vertical migration patterns of mesopelagic organisms, the water column was divided into 10-meter depth bins. Within each bin, spiking signals were normalized by calculating the proportion of spiking points relative to the total number of detected spiking points. Environmental factors were then averaged within each bin. Profiles were further categorized into daytime and nighttime based on local solar time.

For the seasonal analysis, profiles containing spiking layers were categorized, and those with comprehensive environmental data were selected, yielding 1,045 profiles for spring, 1,722 for summer, 1,739 for autumn, and 801 for winter. Following established literature and conventional definitions of depth ranges for the upper (0–200 m) and middle (200–800 m) oceanic zones, the intensity and frequency of spiking signals were quantified for each season. To account for seasonal variations in the number of profiles, frequency distributions were normalized. Frequency, defined as the number of pinnacles per unit depth, represents the likelihood of aggregation for specific taxa. Intensity, measured as the median number of $b_{bp}$ signals within each pinnacle layer, serves as a proxy for species composition (size) or abundance within the mesopelagic zone. For spatial distribution analysis, a grid with a 1° resolution was employed to statistically average the depth distribution of mesopelagic organisms across the study area.

To capture the complex, nonlinear relationships influencing the distribution of mesopelagic organisms, we utilized a Random Forest model, building on methodologies from prior studies(De Forest and Drazen, 2009; Scales et al., 2016; Cuttitta et al., 2018; Villafaña and Rivadeneira, 2018; Song et al., 2022; Alexander et al., 2023). Initially, Spearman correlation analysis was conducted to explore the associations between the depth and intensity of spiking signals within the pinnacle layer and environmental variables. Subsequently, we applied the Random Forest model to elucidate the regression relationships between mesopelagic organism densities and environmental factors, thereby providing a nuanced understanding of their interactions.

## 3   Results

### 3.1   Diurnal Vertical Migration

Our findings reveal a distinct diurnal migration pattern with a multilayered structure in mesopelagic organisms. Normalized data indicate prominent intensity bands within the $b_{bp}$ cusp layer at depths of approximately 150 m, 330 m, 650 m, and 780 m during daylight hours. In contrast, nighttime observations show a strong scattering layer at a shallower depth of around 350 m. Notably, the mean intensity of the $b_{bp}$ spike layer at depths shallower than 380 m is 1.24 times higher at night compared to daytime, increasing to 1.28 times at depths shallower than 100 m. Conversely, at a depth of approximately 380 m, the average daytime intensity is 1.17 times higher than nighttime values (Fig. 3a, b).

These observations are consistent with the acoustic findings by(Klevjer et al., 2020a) across four regions, indicating a substantial overlap in mesopelagic distribution patterns detected by both methods. Our results suggest that spike layer signals predominantly concentrate in mid-ocean layers during the day, with a significant portion migrating to upper ocean layers at night.





**Figure 3.** This figure illustrates the diurnal distribution of $b_{bp}$ signals and environmental factors, with colored lines indicating daytime and grey lines representing nighttime. Specifically, figure a depicts $b_{bp}$ signals, figure b shows chlorophyll levels.

## 3.2 Seasonal Vertical Migration

Seasonal analysis reveals notable variability in the intensity of $b_{bp}$ spike layer signals across different ocean layers. The average intensity between the upper and middle layers exhibits minor differences in spring and autumn (below 10%), whereas more substantial disparities are observed in summer and winter (over 50%). In spring and winter, $b_{bp}$ spike intensity in the upper ocean is generally lower than in the middle layer, with the opposite pattern in summer and autumn. The upper ocean reaches





peak intensity in summer (0.002290 m$^{-1}$) and minimum intensity in winter (0.001635 m$^{-1}$). In the middle layer, peak intensity occurs in winter (0.005748 m$^{-1}$), with the lowest in autumn (0.001502 m$^{-1}$) (Table 1).

**Table 1.** Seasonal average intensity of Mesopelagic organism aggregation in the upper and middle layers of the ocean; The peak intensity denotes the highest intensity recorded for a particular layer, with the depth indicating the precise location .

| season | layer(depth) | intensity(m$^{-1}$) | peak intensity m$^{-1}$(depth) |
|---|---|---|---|
| spring | upper layer(0-200m) | 0.001699 | |
| | middle layer(200-800m) | 0.001760 | 0.003369(510m) |
| summer | upper layer(0-200m) | 0.002290 | 0.003225(20m) |
| | middle layer(200-800m) | 0.001647 | |
| autumn | upper layer(0-200m) | 0.001665 | |
| | middle layer(200-800m) | 0.001502 | 0.002064(440m) |
| winter | upper layer(0-200m) | 0.001635 | |
| | middle layer(200-800m) | 0.002582 | 0.005748(600m) |

The distribution of extreme values in the $b_{bp}$ cusp layer intensity exhibits clear seasonal patterns. In spring, extreme signals
155  appear around 350 m, 510 m, and 700 m, with a peak intensity at 510 m (0.003369 m$^{-1}$). During summer, the highest intensity shifts to the near-surface layer, around 20 m (0.003225 m$^{-1}$). In autumn, intensity is primarily concentrated between 300 m and 600 m, with a peak around 440 m (0.002064 m$^{-1}$). Winter signals concentrate between 200 m and 700 m, with maximum intensity at 600 m (0.005748 m$^{-1}$) (Fig. 4). These findings suggest that the maximum intensity of the $b_{bp}$ cusp layer predominantly occurs in the middle ocean layer during spring and winter, reflecting a multilayer aggregation pattern, while in
160  summer, the highest intensity is near the surface. In autumn, the difference between upper and middle layers is less pronounced, consistent with the findings of (Loisel et al., 2002)for the same region. Additionally, the depth of the strongest $b_{bp}$ spike signal demonstrates a distinct seasonal dynamic: it is deepest in winter (around 600 m), ascends in spring (approximately 510 m), rises further to near-surface levels in summer (around 100 m), and descends in autumn (about 440 m).





**Figure 4.** The diagram depicts the density distribution of mesopelagic organisms in different seasons. The four seasons are represented by a, b, c, and d, corresponding to spring, summer, autumn, and winter, respectively. The shaded areas correspond to the spike layer, while the error bars indicate the standard deviation of the depth range.

The normalized frequency distribution of the $b_{bp}$ spike layer across different seasons (Fig. 5) reveals a consistent trend of
165 relatively high-frequency aggregation of mesopelagic organisms at depths shallower than approximately 350 m. In spring and winter, the average frequency of $b_{bp}$ spike signals within the upper 350 m is elevated by factors of 1.28 and 1.33, respectively, compared to deeper waters. Additionally, there is a significant increase in the proportion of organisms migrating to the upper 200 m during spring, rising from 1.17 in winter to 1.58, indicating a notable shift toward shallower depths. In summer and




autumn, the mean frequency of $b_{bp}$ spike signals at depths shallower than 350 m is 1.85 and 4.15 times higher, respectively,
than at greater depths. Notably, there is a pronounced aggregation of high-frequency signals in the near-surface layer, shallower than 50 m. These observations suggest that in spring and winter, despite lower average $b_{bp}$ spike intensity in the upper ocean compared to the mid-layer—with peak values primarily in the mid-layer—mesopelagic organisms aggregate at specific mid-layer depths while foraging in the upper ocean. In contrast, in summer and autumn, especially summer, both the average intensity and frequency of $b_{bp}$ spikes are significantly higher in the upper layer than in the mid-layer, with a marked concentration in the near-surface zone. This shift indicates a seasonal change in mesopelagic behavior, with a heightened preference for upper-layer habitats and foraging during warmer months.

**Figure 5.** This figure illustrates the intensity and density of mesopelagic organisms vertical distribution, with red areas indicating multiple occurrences of these organisms throughout the mesopelagic zone. The color bar represents the proportion of occurrences of mesopelagic organisms within each 5m bin relative to the total seasonal frequency.



## 3.3 Horizontal Spatial Distribution

The horizontal spatial distribution of mesopelagic organisms was analyzed by calculating the mean depths of all $b_{bp}$ cusp layers within a $1° \times 1°$ grid across mid- and high-latitude regions of the North Atlantic (Fig. 6). The results indicate a predominantly shallow distribution in the northwestern North Atlantic and the Mediterranean Sea, with mean depths around 200 m. In contrast, the Labrador Sea shows a deeper average $b_{bp}$ cusp layer depth of approximately 400 m, while the Irminger Sea averages around 300 m. In the eastern Iceland Sea, organisms are found at substantially greater depths, averaging around 800 m. Prominent frontal zones, including the Greenland-Icelandic-Norwegian Fronts, Eastern Greenland Fronts, La Nerado Fronts, and North Atlantic Drift Fronts, show $b_{bp}$ cusp layers at depths ranging from several tens to a few hundred meters.

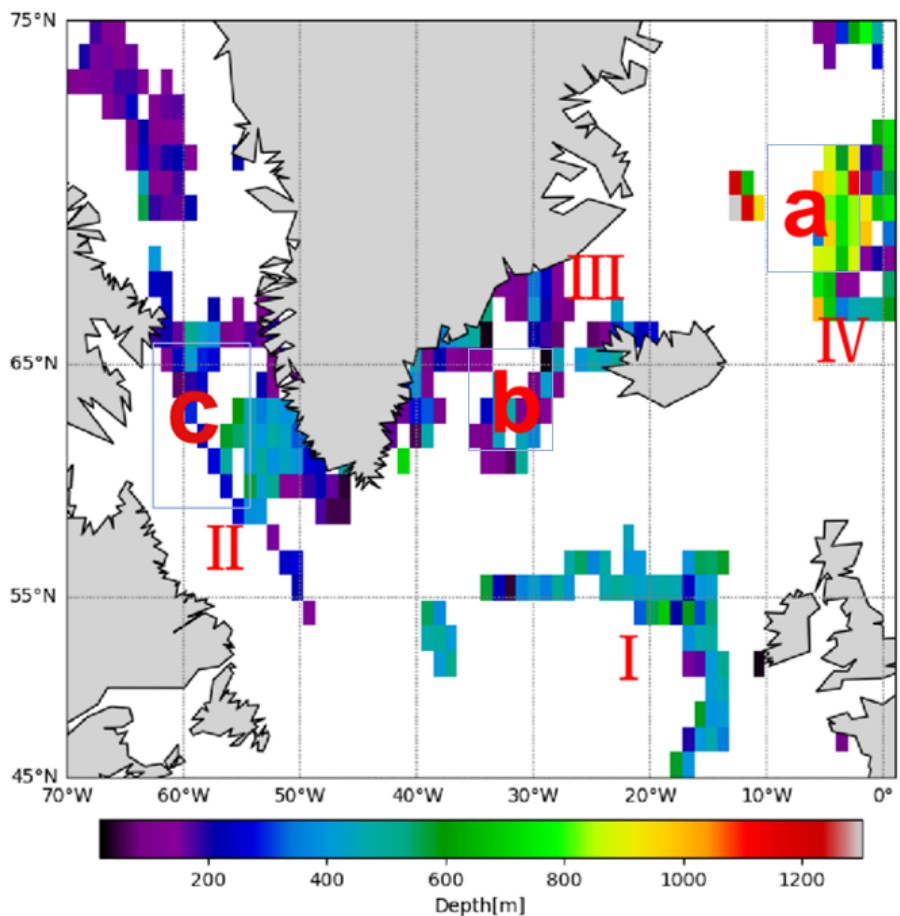

**Figure 6.** The figure shows the spatial differences in the depth distribution of mesopelagic organisms, with a spatial resolution of $1° \times 1°$. The map shows the Iceland Sea (a), Irminger Sea (b), and Labrador Sea (c). The North Atlantic Drift Front (I), Laredo Front (II), East Greenland Front (III), and Greenland-Iceland-Norway Front (IV) are indicated.



## 3.4 Environmental driving factors

Spearman correlation analysis revealed that the vertical distribution of mesopelagic organisms is influenced by several environmental factors, despite the low contribution rates of individual variables. Subsequent analysis using the Random Forest model identified the temperature gradient as the most significant factor, explaining 29.44% of the variance. Other influential factors include seawater salinity in the 500-800 m layer (15.49%), dissolved oxygen in the 500-800 m layer (14.85%), mean sea surface temperature (SST) (14.41%), and sea surface chlorophyll concentration (SSC) (13.46%). Latitude accounted for 12.35% of the variance. The model's overall goodness of fit was represented by an $R^2$ value of 0.58. Among these factors, the temperature gradient exerted the most substantial influence on the seasonal and spatial distribution of mesopelagic organisms. Throughout the year, light conditions (solar altitude angle) were notably influential in spring, while summer distributions were predominantly driven by sea surface chlorophyll concentration. Dissolved oxygen was the most significant factor in autumn and winter. Latitude, as a critical geographic factor, also influenced distribution patterns, with mesopelagic organisms typically found at shallower depths at higher latitudes. Model response curves further elucidated the relationship between environmental factors and mesopelagic aggregation depth. An increase in temperature gradients within a specific range was associated with shallower depths of mesopelagic organism aggregation. This observation partly explains the alignment between the spatial distribution of mesopelagic organisms and temperature heterogeneity. Conversely, when considering the intensity of biotic aggregation as the response variable, signals from mesopelagic organisms at shallower depths were generally more intense.





**Figure 7.** Response curves from the random forest model, with the blue line indicating the influence of various factors. The small black line on the horizontal axis denotes data distribution density, while the gray points represent individual data points.

## 4 Discussion

Recent research underscores the complex factors influencing the aggregation and vertical migration of mesopelagic organisms, driven by a myriad of environmental variables such as food availability, light conditions, oceanographic features, and hypoxia. Short-term transient influences, including variations in cloud cover, ocean currents, and lunar phases, further modulate these

processes(Lampert et al., 1989; Ramos-Jiliberto et al., 2002; Parra et al., 2019; Klevjer et al., 2020b; Hauss et al., 2016).

During the day time, mesopelagic organisms predominantly reside within the mesopelagic zone (150–800 m), forming distinct signal bands. Conversely, nocturnal migrations lead these organisms to occupy shallower pelagic strata, particularly those below 380 m. Despite these observations, the environmental dynamics within these shallower strata remain insufficiently





defined. Comparative analyses indicate that elevated chlorophyll concentrations, more favorable thermal conditions, and re-
duced nocturnal illumination in the upper pelagic layers collectively reduce predation risk and avoid hypoxic conditions. These
factors create a more favorable environment for mesopelagic organisms, thereby enhancing their nocturnal migrations and in-
tensifying the $b_{bp}$ signal in the upper pelagic layers, consistent with satellite-based lidar observations(Behrenfeld et al., 2019).
Additionally, the interaction between diurnal $b_{bp}$ spiking layer characteristics and environmental factors such as chlorophyll
and temperature emphasizes the importance of thermal and salinity gradients. Enhanced spiking signals are observed above
these gradients, driven by increased food availability and physiological predispositions favoring aggregation in regions of
chlorophyll maxima and thermal gradients(Sameoto, 1986). These findings align with Random Forest model results, which
demonstrate that pronounced temperature gradients correlate with shallower mesopelagic distributions.

Seasonal variations in the $b_{bp}$ spike layer intensity are influenced by multiple environmental factors, including water temper-
ature, ocean currents, dissolved oxygen levels, light availability, and food sources(Bianchi et al., 2013; Klevjer et al., 2016). The
impact of these factors varies significantly across different regions and seasons(Klevjer et al., 2020a), leading to fluctuations in
mean intensity, intensity maxima distribution, and the frequency of $b_{bp}$ spike layer signals within the mesocosm. During spring
and winter, the mean intensity of the $b_{bp}$ spike layer in the upper mesopelagic zone decreases, while its frequency increases
relative to the middle mesopelagic layer. This shift likely results from mesopelagic organisms' preference for specific depths
influenced by lower temperatures, deeper mixed layers, limited light availability, and reduced phytoplankton concentrations
in the upper layers during these seasons. In contrast, as temperatures and light levels rise in spring, and with the mixed layer
becoming shallower and phytoplankton blooms increasing, mesopelagic organisms migrate to the upper layers for improved
foraging opportunities. This behavior results in higher-frequency aggregations and a relative decrease in mean intensity of
the mesopelagic acropora signal(Allan et al., 2021; Henson et al., 2012; Lutz et al., 2007; Woodd-Walker et al., 2002; Briggs
et al., 2011; Vedenin et al., 2022). Furthermore, in cooler spring and winter months, strong downwelling increases surface
density, while salinity differences and stratification in high latitudes and the Atlantic Ocean facilitate the transfer of dissolved
oxygen to deeper waters. Consequently, mesopelagic organisms migrate to greater depths in search of suitable habitats and
food resources, avoiding elevated predation pressure in surface waters(Freeman, 2006; Garcia-Soto et al., 2021; Yin et al.,
2024). This results in a higher concentration of organisms in the middle layer, leading to multilayer aggregation phenomena.
The correlation between dissolved oxygen in the 200–500 m layer and the negative correlation in the 500–800 m range indi-
cates a distinct oxygen minimum zone around 500–600 m, delineating the emergence of a prominent mesopelagic signal layer
at approximately 600 m depth. Larger mesopelagic organisms, resistant to currents, migrate deeper, while smaller organisms
remain in the upper layers. These observations align with Random Forest analyses and previous studies showing that smaller
species dominate in surface waters, while larger species are more prevalent in deeper layers(Lin and Costello, 2023; Sorochan
et al., 2023). During spring and winter, the North Atlantic's deeper mixed layer and unstable water column, along with transient
stratification events often disrupted by storms, favor the accumulation of organic matter in the deeper mixed layer, resulting in
increased biotic aggregation frequencies in the mid-ocean(Dall'Olmo et al., 2016).In contrast, during summer, a stable shallow
mixed layer isolates the surface from deeper waters, concentrating mesopelagic organisms in the upper-middle layer. High-
intensity and high-frequency signal layers emerge in the ocean's surface during summer and autumn. In autumn, these strong



signals are frequently associated with chlorophyll maxima around 200 m depth. Increased solar radiation enhances phyto-
plankton photosynthesis, significantly boosting primary productivity and providing abundant food resources for larger marine
organisms(Flombaum et al., 2013). Warmer sea surface temperatures also create favorable conditions for species thriving in
warmer waters, promoting the survival, reproduction, and growth of larger marine organisms(Chen et al., 2019; Bova et al.,
2021). Additionally, ocean circulation and upwelling transport nutrient-rich deep waters to the surface, attracting larger marine
species to feed during the day. The mesopelagic layer at 400–500 m, typically inhabited by non-swimming species or crus-
taceans, is shaped by vertical fluxes of organic carbon and particulate matter(Marohn et al., 2021; Liu, 2011; Sikder et al.,
2019; Henson et al., 2012; Lutz et al., 2007). These factors collectively contribute to the aggregation of mesopelagic organisms
at the sea surface during summer and autumn. Similar patterns are observed in the Gulf of Mexico, where high stratification in
summer and autumn leads to high signal abundance and frequency in the surface layer, whereas spring and winter show deeper
mixed layers and predominantly deeper distributions(Contreras-Catala et al., 2016), reflecting high mesopelagic activity in the
upper ocean layer and strong aggregation during these seasons.

The spatial distribution of mesopelagic organisms reveals a pattern of shallower depths in warmer regions, gradually deep-
ening with increasing latitude. This latitudinal shift is primarily attributed to variations in light conditions, consistent with prior
acoustic studies that report a positive linear correlation between deep scattering layer (DSL) backscattering and temperature
at corresponding depths(Proud et al., 2017, p = 0.0001). In the Iceland Sea, carbon flux to the upper ocean is more than eight
times lower than in lower-latitude regions, leading to a general mid-ocean distribution of mesopelagic organisms(Klevjer et al.,
2020a). However, latitude alone does not fully explain the spatial distribution differences observed. For example, despite sim-
ilar latitudes, the western side of Greenland's Iceland Sea hosts shallower mesopelagic populations compared to the eastern
side, with the reasons for this disparity remaining unexplored. Given the North Atlantic's complexity, factors such as sea ice
cover, the North Atlantic Oscillation, and diverse current systems likely influence mesopelagic distributions(Gu et al., 2024;
Puerta et al., 2020; Lynch-Stieglitz et al., 2024). Environmental differences show that the eastern coast of Greenland, influ-
enced by the North Atlantic Warm Current, experiences higher temperatures and increased chlorophyll concentrations, creating
more favorable conditions for mesopelagic organisms, predominantly found in mid-ocean layers. In contrast, the western coast,
shaped by the Greenland Cold Current, features lower temperatures and reduced nutrient availability. The colder sea surface
on the west may decrease large predator activity, providing a relatively safer habitat with suitable nutritional conditions for
mesopelagic organisms(Chawarski et al., 2022).

Mesopelagic organisms also exhibit significant aggregation behaviors in frontal zones, where alternating downwelling and
upwelling currents induce vertical displacements with substantial ecological impacts. These regions are characterized by steep
environmental gradients, including variations in sea surface temperature, chlorophyll concentration, sea surface height, and dis-
solved oxygen, which significantly influence fish distribution(Owen, 1981; Woodd-Walker et al., 2002). Frontal zones, marked
by distinct thermal gradients resulting from the convergence of different water masses, serve as biodiversity and productivity
hotspots(Longhurst, 2007). In the study area, an average of three to four water masses interact, with polar fronts demarcating the
boundary between Atlantic and polar water masses. The interaction between colder northern waters and terrestrial runoff cre-
ates gradients of declining temperature and salinity, forming distinct physiographic environments that influence mesopelagic



distribution(Sutton et al., 2017; Astthorsson et al., 2007). Notably, the Greenland-Iceland-Norway front, characterized by a

significant temperature-salinity gradient, corresponds to deeper mesopelagic aggregations, driven by the separation of colder Arctic and warmer Atlantic waters and the resulting temperature-salinity gradient, which critically impacts marine productivity and spatial distribution. Moreover, vertical mixing within frontal zones enhances nutrient upwelling, supporting higher primary productivity and providing abundant food resources for mesopelagic organisms(Ljungström et al., 2021). Previous studies have similarly highlighted the role of water masses and frontal zones in influencing mesopelagic distributions(Yin et al., 2024).

**5 Conclusions**

Comparative analysis using acoustic trawl and satellite lidar detection confirms that $b_{bp}$ from BGC-Argo effectively capture the biological signal of mesopelagic organisms. During the day, mesopelagic organisms predominantly inhabit the middle layers, exhibiting multi-layered aggregation patterns. At night, reduced light levels lower predation risks, driving a general upward migration into the upper layers, where pronounced diel vertical migration (DVM) is observed.

Seasonally, the mean intensity of $b_{bp}$ spikes in the upper layers remains lower than in the middle layers during spring and winter, although the frequency of these spikes in the upper layers is higher. In contrast, summer and autumn show an increase in both intensity and frequency of $b_{bp}$ signals in the upper layers, particularly near the surface. This seasonal shift reflects a change in habitat utilization, with mesopelagic organisms becoming more active in the upper layers for foraging. The depth of the strongest $b_{bp}$ signal exhibits a periodic pattern, shallowing from winter through spring and summer, and deepening in

autumn, which corresponds to seasonal fluctuations in mixed layer concentration.

Horizontally, the study area reveals deeper distributions in the northeast and shallower distributions in the southwest. In the northwestern North Atlantic, mesopelagic organisms typically reside at an average depth of 200 meters, while in the eastern Iceland Sea, they are found at greater depths, around 800 meters. The Labrador Sea features an average signal layer depth of 400 meters, whereas the Irminger Sea has it at approximately 300 meters. Oceanic fronts, such as the Greenland-

Iceland-Norway Front, East Greenland Front, Labrador Front, and North Atlantic Drift Front, present pronounced temperature gradients, favorable light conditions, and nutrient-rich waters, attracting significant concentrations of mesopelagic organisms and leading to substantial biological aggregations.

Spatiotemporal distribution patterns highlight that mesopelagic depth distribution is influenced by multiple environmental factors. Correlation and random forest analyses underscore temperature as a primary determinant year-round, with temperature

gradients emerging as the most significant factor affecting mesopelagic distribution in peak layers. Seawater salinity, dissolved oxygen, sea surface chlorophyll concentration, and latitude also play important roles.

BGC-Argo data provide valuable insights into the spatial distribution and seasonal variability of mesopelagic organisms, promoting our understanding of organic carbon transfer to the deep sea, ecosystem energy and material cycling, and fisheries management. Future research should incorporate additional environmental factors such as eddies, currents, and oceanic fronts

to further elucidate the complex dynamics influencing mesopelagic organisms. Despite the extensive vertical profile data pro-



vided by BGC-Argo, clustering effects and limited sampling of certain environmental parameters suggest that advancements in lidar technology could substantially improve mesopelagic organism detection capabilities.

*Data availability.* BGC-Argo data can be accessed through the Biogeochemical-Argo portal (https://biogeochemical-argo.org/data-access.php). Other datasets, such as GlobColour Sea surface chlorophyll (CHL) data (https://www.globcolour.info) and Sea surface temperature data
(https://www.ncei.noaa.gov/metadata/geoportal/rest/metadata/item/gov.noaa.ncdc:C00844/html, doi:10.7289/V5SQ8XB5), are also available for download.

*Author contributions.* Jie Yang and Jianhui Li contributed equally to this manuscript. Jie Yang was responsible for data collection, funding acquisition, equipment provision, and refining the manuscript's logical structure. Jianhui Li conducted the experiments, obtained the experimental results, completed the experimental methods, and drafted the manuscript. Both authors are co-first authors.

*Competing interests.* The contact author has declared that none of the authors has any competing interests.

ther geographical representation in this paper. While Copernicus Publications makes every effort to include appropriate place names, the final responsibility lies with the authors.

*Acknowledgements.* The author thanks Haëntjens for providing the code for the spike identification algorithm and appreciates the contri-
butions of previous researchers in biogeochemical identification in the ocean. Data from the Biogeochemical-Argo portal, GlobColour Sea surface chlorophyll, and Sea surface temperature datasets were invaluable for this work.



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
