# Peer review of "Temporal and Spatial Influences of Environmental Factors on the Distribution of Mesopelagic organisms in the North Atlantic Ocean"

_EGUsphere, 2024_

## Author Response (AR1)

**Point to point reply**

Dear editor and reviewers,

We greatly thank you for the thorough review of the manuscript and the valuable comments. We have gone through these comments and suggestions carefully, and made revisions based on these comments and suggestions. Our responses are shown below. Due to the possibility of receiving further data modification comments, we did not upload the dataset this time. For your convenience in reviewing the manuscript, we have submitted a PDF file along with the .TeX file. Main changes are highlighted in YELLOW in the PDF file and the details in response to the comments are given below.

AR: Author responses.

"Italic" represents the corresponding changes in the manuscript.

**Reviewer: 1**

Comments to the Author

I have a few major concerns: The methods need more detail. The Material and methods part of the study is very brief which makes it hard to assess whether the used methods are appropriate or not. For example:

AR: Thank you for your insightful review and suggestions. We have revised the Materials and Methods section for greater clarity, providing more detailed descriptions and justifications of the methodologies.

Comments 1:Why did the authors include remotely sensed data of variables that are measured by BGC-Argo anyways? Wouldn't the in situ data be more accurate and surely match the location of the floats compared to 4 km resolution satellite data?

AR: Thank you for your comment. While BGC-Argo data provide high accuracy at float locations, satellite-derived data, such as Sea Surface Temperature (SST) and Chlorophyll-a (Chl), offer broader spatial and temporal coverage, particularly in areas where in situ measurements are limited. We have improved the relevant description as follow.

"Satellite-derived parameters like SST and Chl provide context for tracking surface ocean dynamics influencing mesopelagic distributions and examining large-scale seasonal and annual trends. They also establish temporal baselines and environmental context for BGC-Argo data, particularly in regions with limited in situ measurements or where large-scale trends are assessed."

Comments 2:The calculation of the smad needs to be better explained with a description of all the terms included in equation (1).

$$smad = -\frac{1}{\sqrt{2} \bullet erfcinv(\frac{3}{2})} \bullet median(|b_{bp}(i) - median(b_{bp}(n))|)$$

AR: Thank you for your suggestion. We have improved the relevant description as follow.

"where smad represents the minimum threshold of the profile, defined as the standardized median absolute deviation of the signal distribution. bbp(i) represents each backscattering coefficient bbp value in the profile, while bbp(n) refers to the set of all bbp values in the profile. The calculation of the median is performed on the deviations of all spike values from the median of spikes in the profile. The erfcinv(3/2) term, the inverse complementary error function evaluated at 3/2, serves as a scaling factor for standardizing smad."

Comments 3:Why did the authors calculate vertical temperature gradients? How do they expect they would impact the distribution of mesopelagic organisms? Is this an assessment to measure stratification (e.g., Fernandez et al., 2017)? If so, why not using MLD instead? Or are they horizontal gradients?

AR: Thank you for your valuable comments. We calculated vertical temperature gradients as proxies for stratification, which influence the vertical distribution and

migration of mesopelagic organisms. Previous studies (e.g., Proud et al., 2017) show a strong correlation between backscattering intensity and temperature at mesopelagic scattering layers (DSL). We matched these gradients to mesopelagic organism locations to explore this relationship. Following your recommendation, we also used the hybrid algorithm (Holte et al., 2017) for more accurate mixed layer depth (MLD) estimates. We have improved the relevant description as follow.

"In addition to these key parameters, we incorporated two additional variables to enhance our analysis: Photosynthetically Active Radiation (PAR) and Mixed Layer Depth (MLD). These variables provide important insights into light conditions and the vertical structure of the ocean, both of which are critical for understanding the dynamics of mesopelagic organisms."

--Line 106-109 in highlight version

"For MLD, we used data from the hybrid algorithm and threshold method (Holte et al., 2017). Among them, the hybrid algorithm was preferred for its accuracy, especially in regions like the Labrador and Irminger Seas, where the threshold method overestimates MLD by 10% in winter."

--Line 112-114 in highlight version

Comments 4:In Figure 2 - what are the profiles 'Profile 1' and 'Profile 2' that are used in the analysis? How can they be concurrent?

AR: Thank you for your valuable comment. Profiles "Profile 1" and "Profile 2" are considered concurrent if their depth ranges overlap and the overlap count meets a predefined threshold. The determination process is as follows:

**1. Input data format:**

Profiles are provided as a cell array, where each element contains the features of a profile, such as p\_shallow (shallow boundary) and p\_deep (deep boundary). The format of each profile matches the structure of c1\_features in our analysis.

**2. Pairwise comparison:**

For any two profiles (p1 and p2), we check their overlap in depth ranges:

Part 1: Determine whether p2 data points fall within the range of p1 (p\_shallow to p\_deep).

Part 2: Determine whether p1 data points fall within the range of p2. The total number of overlapping events between the two profiles is recorded in the overlap count matrix (alpha matrix).

**3. Concurrency determination:**

If the overlap count between two profiles meets or exceeds the specified threshold (THRESHOLD), the profiles are considered concurrent. This concurrency is stored as a logical value in the concurrency matrix (qc matrix).

"Figure 2: Extraction process of spike layer signal. Profiles "Profile 1" and "Profile 2" are considered concurrent if their depth ranges overlap and the overlap count exceeds a predefined threshold."

--Line 144-145 in highlight version

Comments 5:The authors mentioned dividing profiles by whether they are from the day or night. What about profiles collected at dawn and dusk? Most BGC-Argo profiles will likely measure profiles around noon or midnight so I suspect there won't be many of those, but profiles at dawn and dusk could confuse the analysis so I believe should be excluded or treated separately.

AR: Thank you for your suggestion. We examined the time distribution of BGC-Argo profiles and found that dawn and dusk measurements are sparse. In line with your recommendation, we have excluded the two-hour periods around dawn and dusk and updated the day-night distribution plot accordingly. This adjustment has been made to improve the clarity and reliability of the analysis.

Comments 6:The random forest model needs to be described in more detail. For example, what is the response variable that is then depicted in Figure 7- is it the anomaly of the depth of the main spike? Also, how was the random forest parameterized (e.g., number of trees, etc.?)

AR: Thank you for your suggestions regarding the random forest model. We have revised the manuscript to include a more detailed description, specifying the response variable and the model parameterization. The relevant changes to the manuscript are provided below.

"Figure 7. Response curves from the random forest model, with the blue line indicating the influence of various environmental factors. The small black ticks along the horizontal axis represent the distribution density of the data, while the gray points represent individual data points. The X-axis displays the range of feature values. The Y-axis shows the accumulated local effect (ALE) of each feature on the response variable (p), which reflects the anomaly in depth change of the primary spike layer. Positive values indicate a deepening of the spike, while negative values indicate a shoaling."

--Line 243-245 in highlight version

"The Random Forest model was parameterized with 500 trees (ntree), balancing performance and computational efficiency. We used the default number of variables per split (mtry), where the value of mtry was set to the square root of the total number of input features. This configuration allowed the model to capture intricate, non-linear patterns in the data. The model exhibited robustness in handling high-dimensional data, achieving an  $R^2$  value of 0.64, indicating moderate explanatory power without signs of overfitting."

--Line 167-172 in highlight version

"Random Forest variable importance analysis revealed that the vertical temperature gradient made the greatest contribution to the model, accounting for 26.03% of the variance. Following this, latitude (13.92%), dissolved oxygen at 500 m (13.71%), photosynthetically active radiation (PAR, 8.66%), salinity at 500 m (8.29%), mixed layer depth (MLD, 8.23%), average chlorophyll concentration (8.09%), average temperature (7.10%), and solar altitude (6.68%) were identified as the next most important factors. Among these, the vertical temperature gradient had the most significant impact on the seasonal and spatial distribution of mesopelagic organisms. Latitude, as a key geographical factor, also exerted a considerable influence on the spatial distribution patterns. Excluding the northeastern regions, mesopelagic organisms were generally found at shallower depths in higher latitudes. The model's response curves further elucidated the relationships between environmental factors and the aggregation depth of mesopelagic organisms in the open ocean. Within certain ranges, increasing latitude, higher dissolved oxygen levels, greater mixing, reduced light penetration, and decreasing temperatures all corresponded to shallower aggregation depths for midwater organisms. Across all regions, the distributions in summer and autumn tended to be shallower, whereas spring and winter distributions were generally deeper. These observations partially explain the consistency between the spatial distribution of midwater organisms and the heterogeneity of the physiological environment. In contrast, when considering the intensity of biological aggregation as a response variable, stronger signals from mesopelagic organisms typically originated from shallower depths. It is important to note that while Random Forest analysis can capture broad trends within specific ranges of environmental variability, the detailed seasonal differences across individual subregions require further multi-factorial analysis for a more comprehensive understanding."

--Line 226-243 in highlight version

"Other critical factors include latitude, dissolved oxygen, par, salinity, mld and surface chlorophyll concentration,, with relative importance of 26.03%, 13.92%, 13.71%, 8.66%, 8.29%, 8.23% and 8.09%, respectively."

--Line 16-18 in highlight version

Comments 7:The authors mention (and I think it's a good idea) normalised profiles. Yet, I couldn't find a very clear explanation of how profiles are normalised (including in the Figure 2 diagram). Please explain this important step of your analysis.

**AR:** Thank you for your thoughtful suggestion regarding the normalization of profiles. We appreciate your feedback and will provide a more detailed explanation of this important step in our analysis.

"To account for seasonal variations in both the number of profiles and overall signal strength, both frequency and intensity distributions were normalized. Specifically, the frequency and intensity in each 10-meter depth bins were normalized by dividing the total number of spike points and the signal strength, respectively, by the total number of detected spike points across the total number of profiles for each season. This normalization procedure minimized potential biases arising from seasonal differences in sampling effort or signal intensity, allowing for meaningful comparisons of vertical distribution patterns across seasons."

--Line 156-161 in highlight version

Some of the results appear to be a bit confusing to me. In particular:

Comments 1: Something that I find challenging is how to disentangle the spatial variability from the temporal one. The area of study includes regions with complex and diverse oceanographic regimes (e.g., Della Penna and Gaube, 2019 for differences within the western side of the domain, but also strong differences between

the eastern and western side) so if the profiles for a specific season are mostly from a specific region there is a risk of associating the typical pattern observed with the season rather than the region. There are a few extra analyses that could help with this, for example: Figure 1. What is not clear to me is - are these the used validated profiles or the total profiles available for the regions? Why is this picture so different from Figure 6?. I recommend including in the supplementary material the equivalent map plotted for each individual season to identify if there is a spatial bias associated with each season.

AR: Thank you for your insightful comments on disentangling spatial and temporal variability. We clarify that Figure 1 shows the distribution of all BGC-Argo profiles with the bbp parameter, while Figure 6 represents the spatial distribution of bbp spike layers extracted from these profiles, explaining the numerical differences. In response to your suggestion, we have included seasonal maps of profile distributions in the supplementary material. These maps reveal slight regional variations across seasons, but overall spatial patterns remain consistent, with shallower distributions in the southwest and deeper ones in the northeast, indicating limited seasonal influence.

Comments 2: Some results are described in the text but are not clearly backed up by figures or tables. For example, in lines 193-195 the authors talk about light conditions, but I wonder if these are included in the modelling? Even just having the solar angle could be a way to incorporate season and location in a single descriptor.

**AR:** Thank you for highlighting the importance of incorporating light conditions into our analysis. We've added PAR to our data and updated the random forest response curves accordingly.

"In addition to these key parameters, we incorporated two additional variables to enhance our analysis: Photosynthetically Active Radiation (PAR) and Mixed Layer Depth (MLD). These variables provide important insights into light conditions and the vertical structure of the ocean, both of which are critical for understanding the dynamics of mesopelagic organisms. For PAR, we utilized a high-resolution, long-term global gridded PAR product (2010–2018) provided by Tang (2021), which has a temporal resolution of three hours. Unlike solar altitude, which is based on latitude and time and may not fully capture the temporal and spatial variability in PAR, this dataset offers a more accurate and detailed representation of light availability."

--Line 106-112 in highlight version

Comments 3: Figure 7 is from my perspective where a lot of valuable results should be but it's not as clear as I think it should be: what are the explained variables on the y axis (e.g., what are the numbers -150,200 meaning on the axis?) and it doesn't include any units.

**AR**: Thank you for your comments. The y-axis in Figure 7 represents the anomaly in depth change of the primary spike layer. We have updated the figure caption accordingly.

"Figure 7. Response curves from the random forest model, with the blue line indicating the influence of various environmental factors. The small black ticks along the horizontal axis represent the distribution density of the data, while the gray points represent individual data points. The Y-axis shows the accumulated local effect (ALE) of each feature on the response variable (p), which reflects the anomaly in depth change of the primary spike layer. Positive values indicate a deepening of the spike, while negative values indicate a shoaling. The X-axis displays the range of feature values."

--Line 244 in highlight version

The interpretation feels a bit overstretched to me in a few points:

Comments 1: The authors compare the patterns they observed with those identified by Klevjer et al., 2020a, yet both the Klevjer et al., papers in 2020 are focused on a small portion of the domain of this study. I suggest the authors be cautious with how they are connecting their findings with the Klevjer et al., 2020 papers and potentially use other references to compare their findings with. For example, Klevjer et al., 2016 show results from the southern part of the North Atlantic and Della Penna and Gaube, 2020, Wiebe et al., 2023, and Fennell and Rose, 2015, show results from the Western side of the North Atlantic. I'm sure there are more studies the authors could consider to compare the distributions of spike layers they observed.

**AR:** Thank you for your valuable comments. Based on your suggestions, we have expanded the literature review to include additional relevant studies, providing a more comprehensive regional context to support our findings.

"These patterns align with observations by (Klevjer et al., 2016) in the southern North Atlantic, where mesopelagic organisms exhibited significant aggregation between 400–600 meters after dawn, followed by a substantial migration to the upper layers (0-200 m) after dusk. In addition, (Grimaldo et al., 2020) reported three distinct sound scattering layers (SSLs) between 46°-50°N and 21°-26°W, with layers observed at 100-250 m, 300-360 m, and 420-700 m during daylight hours. These findings correspond with our observations, where mesopelagic organisms' backscatter during the day is predominantly concentrated in the mesopelagic layers. This is further supported by (Fennell and Rose., 2015), who found higher Deep Scattering Layer (DSL) densities in years with increased sea temperatures at the depths of major DSL concentration (400–600 m) in the western North Atlantic. Further, (Klevjer et al., 2020), in their study of the Irminger Sea, located northeast of our study area, observed a weak, non-migrating layer at approximately 700 m. This depth coincides with the lower edge of the scattering layer observed in the northeastern region of our study area, providing additional context for the consistency of our results across neighboring regions in the North Atlantic. "

--Line 181-191 in highlight version

Comments 2: A few pretty important statements are not backed up by references. For example the statement in lines 209-210 needs references. In addition, I think that in some cases the references used are not really backing up the statements made in the discussion. For example, the Contrerar-Catala et al., 2016 paper to my knowledge deals with the larvae of mesopelagic fish and not mesopelagic fish in general (I'm

also not sure they used the same metrics of signal abundance and frequency so it's hard to make a comparison).

**AR:** Thank you for your insightful comments. We have replaced the reference to Contreras-Catala et al. (2016) with more relevant studies to better support our results and have further analyzed the findings.

"Our analysis of bbp spike signal frequency and intensity reveals significant seasonal differences between the upper and middle layers of the ocean. In spring and winter, although the average bbp spike intensity in the upper ocean is lower than in the middle layer (where spike values are primarily distributed), mesopelagic organisms still aggregate at specific depths in the middle layer and migrate to the upper ocean for foraging. In contrast, in summer and autumn, especially summer, both the average intensity and frequency of bbp spikes are significantly higher in the upper layer than in the middle layer, with a marked concentration in the near-surface zone. This shift indicates a seasonal change in mesopelagic behavior, with a heightened preference for upper-layer habitats and foraging during warmer months. A similar pattern in the mesopelagic scatterers of intermediate to deep layers was noted by (Powell and Ohman 2015), who investigated the scattering characteristics of migratory and non-migratory zooplankton in frontal regions. Their study found that shallower migratory layers, which consist of smaller but more abundant scatterers, are more homogeneously distributed at finer scales. In contrast, deeper non-migratory layers likely consist of fewer but larger scatterers, and these are associated with a lower abundance of organisms, which are likely non-migratory in nature."

--Line 306-320 in highlight version

Comments 3:The relationship between latitude, light levels, and temperature is not explored very clearly. In line 256 the authors discuss the fact that the bbp spikes appear shallower in warmer regions, gradually deepening with increasing latitudes. They attribute this to a change in light conditions, but then they back their statement up with an explanation based on temperature. I think the mechanisms associated with light levels and temperature need to be disentangled (or discussed more clearly at least). To my knowledge, underwater light levels are strongly related with the distribution of DSL with a variability that occurs at scales ranging from the basins (Asknes et al., 2017) to the small differences in cloud coverage (Omand et al., 2021). In general, many species seem to 'follow an isolume' (see example from Della Penna et al., 2022 or Asknes et al., 2017). According to this framework, we expect the

animals inhabiting DSL to be deeper in clearer waters and shallower in waters with higher light attenuation coefficients (see for example Braun et al., 2023). This is quite the opposite of what the authors are describing in their observations. Perhaps there is another mechanism that is dominating here. Could it be a different mesopelagic community (see the work by Proud et al., 2017 or the recent paper by Chawarski et al., 2022). I'm also wondering, could this pattern be explained by the larger abundance of sinking aggregates at high latitudes (smaller phytoplankton could dominate the lower latitudes with particles that just don't make it that deep)?

AR: Thank you for your thoughtful feedback. We appreciate your suggestion to disentangle the mechanisms associated with light levels and temperature in relation to the distribution of mesopelagic organisms. Our study focuses on a high-latitude region, which is unique compared to the areas covered in previous studies such as Asknes et al. (2017). While their work emphasizes the dominant role of light in shaping mesopelagic distributions globally, their study area extends only up to approximately 40°N latitude. In contrast, our study covers regions that follow a different mechanism: as latitude increases, dissolved oxygen levels rise, light penetration decreases, and temperature diminishes, resulting in a shallower distribution of mesopelagic organisms. This trend is particularly pronounced in the high-latitude North Atlantic, where organisms appear to favor areas with higher dissolved oxygen, despite the increased predation risk associated with shallower distributions in high-latitude DSLs. We also recognize the importance of the variability in distribution observed at similar latitudes, particularly in the case of Greenland's western coastline. Even within the same latitudinal range, we observe significant spatial variation in mesopelagic distributions, suggesting that simple latitude-based patterns may be misleading. This complexity underscores the need for more detailed analysis of different regions within the study area. We plan to revise our manuscript to more clearly discuss these regional variations, considering factors such as sea ice coverage, the North Atlantic Oscillation, and the influence of various oceanic currents (Gu et al., 2024; Puerta et al., 2020; Lynch-Stieglitz et al., 2024).

In addition, we acknowledge that the bbp spike signals we analyzed include not only zooplankton but also other potential contributors, such as sinking aggregates and particulate matter. While FDOM appears to be less sensitive to these aggregates, our analysis was limited by the available data. We have conducted separate analyses of FDOM; however, due to the small sample size, the results were inconclusive. As Haëntjens et al. (2020) point out, the spike extraction algorithm used in our study may not fully capture the overall carbon transport by mesopelagic organisms, with a known precision of >90%. Nonetheless, some spikes are still missed, particularly those from sinking aggregates, which could explain some of the discrepancies in the observed patterns. We hope these clarifications address your concerns, and we will

update the manuscript to better explain the complex interplay of light, temperature, and other environmental factors in mesopelagic distributions across latitudes.

"Spatially, our findings on the spatial distribution of mesopelagic organisms align well with Klevjer's study of four North Atlantic basins, with the shallowest distributions around 200m in the Labrador Sea (LS Sea) and the deepest at 500-600m Icelandic Sea (ICS) (Klevjer et al., 2020). Our study area is situated in a high-latitude region, and with the exception of the unique Norwegian Sea area, the distribution of mesopelagic organisms follows a different mechanism across other regions. As latitude increases, dissolved oxygen levels rise, light penetration diminishes, and temperatures decline. In this context, mesopelagic organisms tend to aggregate at shallower depths. This behavior indicates that, despite the higher predation risk associated with the shallower distribution of the deep scattering layer (DSL) in the North Atlantic's high - latitude regions, these organisms still prefer areas with richer dissolved oxygen. The general depth distribution in the northeastern part of our study area is much deeper, whereas the distribution of mesopelagic organisms along the left coastline of Greenland at the same latitude is much shallower. Even at the same latitude, there is considerable variability in the depth distribution, and, therefore, it is misleading to directly infer that mesopelagic organisms become shallower with increasing latitude. Considering the complexity of the North Atlantic, factors such as sea ice coverage, the North Atlantic Oscillation, and various current systems could influence the distribution of mid-water organisms (Gu et al., 2024; Puerta et al., 2020; Lynch-Stieglitz et al., 2024), highlighting the need to address different regions separately."

--Line 311-324 in highlight version

For the distribution of mesopelagic organisms, a hypothesis suggests that due to

the extreme light climate in high-latitude areas, the foraging conditions are poor, limiting the success of mesopelagic fish in these environments. The persistent daylight in summer limits safe foraging in the upper layers during "nighttime," while continuous darkness in winter may restrict visual foraging at any time of day (Kaartvedt, 2008). Therefore, we hypothesize that seasonal differences in our results are primarily driven by light conditions, but latitude-driven distribution differences cannot be fully explained by light alone. While it is theoretically expected that the light comfort zone remains consistent across oceans with varying levels of light penetration, Aksnes et al. (2009) highlight that oxygen-poor waters, in contrast to oxygen-rich waters, exhibit reduced light penetration. The mechanism linking light attenuation to dissolved oxygen may involve microbial heterotrophic degradation of particulate organic matter, leading to the release of CDOM, which exacerbates light attenuation in oxygen-deprived waters (Aksnes et al., 2009; Nelson et al., 2013; Catala et al., 2013). From a biological distribution perspective, our results challenge the general assumption that mesopelagic organisms tend to inhabit deeper layers in clearer waters and shallower layers in waters with higher light attenuation

coefficients (Braun et al., 2023). In high-latitude regions, we observe that mesopelagic organisms tend to distribute shallower, which contradicts the expected pattern where light attenuation should correlate with deeper distributions. This discrepancy may be linked to CDOM peaks associated with zooplankton foraging and excretion behavior, producing fluorescent proteins or amino acid-like fluorescence. This is fundamentally different from the mesopelagic bbp spike signals we detected, which reflect aggregates of zooplankton or sinking materials. Therefore, in high-latitude regions, the latitude-driven distribution of zooplankton or sinking material aggregates is not solely influenced by light conditions. Environmental differences also suggest that the western coast, influenced by the Greenland cold water current, has lower temperatures and reduced nutrient availability. The colder sea temperatures may reduce the activity of large predators, providing relatively safe habitats and suitable nutrient conditions for mesopelagic organisms (Chawarski et al., 2022)." Previous studies (e.g., Kaartvedt, 2008) have suggested that the light climate in high latitudes limits the northward extension of larger mesopelagic fish populations, as both summer light nights and winter darkness limit food availability, in the ICS, migration into the epipelagic zone is restricted by nocturnal light levels. (Norheim et al., 2016).".Langbehn (2022) found that in high latitudes, light conditions primarily regulate the distribution and population dynamics of mesopelagic fish, with temperature playing a secondary role. In winter, as daylight diminishes, prey disperses, and most organisms remain dormant in deeper waters. Cold temperatures and low metabolic demands enable mesopelagic fish to conserve energy despite limited food availability. In summer, warmer temperatures and longer daylight hours force mesopelagic fish to forage near the surface, but increased predation risk drives them to venture outside the optimal light zone in search of food (Langbehn et al., 2022). "Our results also indicate a clear trend of deeper biological distributions in spring and winter, which is similar to the long overwintering phase of squid species that feed and reproduce in deeper waters (Berge et al., 2012).

--Line 328-358 in highlight version

In polar regions, ocean ecosystems are heavily influenced by seasonal changes in light and temperature (Smetacek and Nicol 2005). While light plays a crucial role in the vertical migration of zooplankton and fish, affecting their predation and survival (Kaartvedt, 2008; Ljungström et al., 2021), temperature directly affects physiological rates (Gillooly et al., 2001). Our study region is influenced by polar water masses, acoustic and oceanographic measurements, several studies have demonstrated that latitude-driven variations in upper-layer communities align with the polar boundary defined by deep-sea temperature gradients (Saupe et al., 2019; Sallée et al., 2021). As mesopelagic organisms transition into polar water masses, the acoustic backscattering of these organisms suddenly weakens, and vertical scattering increases, altering the structure of the mesopelagic zone (Ingvaldsen et al., 2023)." In conclusion, our findings demonstrate that light is the primary driver of the seasonal

distribution of mesopelagic organisms in the study area, particularly in high-latitude regions, whereas vertical temperature gradients govern their vertical distribution.

**Minor points**

Throughout the text: I suggest having a space between text and references. For example, at the end of page 1: "nutrient cycling(Klevjer et al., 2016)" would be more readable if written as "nutrient cycling (Klevjer et al., 2016)"

AR: Thank you for your suggestion. Since this issue appears frequently throughout the text, we'll go ahead and make the changes throughout the entire document, without listing each one individually.

Title: I suggest editing the title adding an 's' to 'organism' as I think the authors are interested in more than a specific one.

"Temporal and Spatial Influences of Environmental Factors on the Distribution of Mesopelagic organisms in the North Atlantic Ocean"

Abstract:

Line 5: "spiking layer signals": this is not a concept that is obvious - I suggest using a more commonly used term, maybe 'bbp spikes'?

"This extensive dataset enabled the identification of bbp spikes, allowing us to investigate the diurnal and seasonal vertical distributions of mesopelagic organisms, as indicated by these bbp spikes."

--Line 5-6 in highlight version

Line 14: 'shallower distributions' of what?

"Spatially, mesopelagic organisms migrate deeper in the northeast and remain shallower in the southwest, correlating with higher temperatures and shallower distributions of mesopelagic organisms."

--Line 13-14 in highlight version

Line 16: Please specify that you are talking about vertical temperature gradient (is this the case?)

AR: Yes, that is correct. We are discussing the vertical temperature gradient.

Introduction

Line 25: "mesopelagic zone is diel vertical" -> "mesopelagic zone is the diel vertical"

"A prominent behavioral adaptation in the mesopelagic zone is the diel vertical migration (DVM), wherein organisms undertake extensive vertical movements to optimize survival and foraging efficiency."

--Line 26-27 in highlight version

Line 30: The statement about the SVM needs a reference or two to back it up.

"Additionally, mesopelagic organisms undergo seasonal vertical migration (SVM), adjusting their vertical distribution in response to environmental fluctuations. (Robinson et al., 2010)."

--Line 30-31 in highlight version

Line 40: Why is there a specific reference to ADCP? The statement made here is about all active acoustics approaches (ADCP but also scientific echosounders).

"Additionally, the resolution of acoustic sensors often fails to detect small, dispersed, and weakly scattering species at depth, and the high costs associated with traditional active acoustic methods, including ADCP and scientific echosounders, further limit extensive in situ observations (Haëntjens et al., 2020; Chai et al., 2020; Underwood et al., 2020; Nakao et al., 2021)."

--Line 40-43 in highlight version

Line 46: "renderiimportanceerful" is not a word. Please edit.

"Bio-optical sensors mounted on these floats have proven effective in detecting a range of bio-optical properties, rendering them powerful tools for large-scale spatial detection of mesopelagic organisms (Claustre et al., 2019; Haëntjens et al., 2020)."

--Line 47-49 in highlight version

Line 56: In what way the Behrenfeld et al., 2019 paper discussed evolutionary patterns?

AR: Thank you for your comment, and we apologize for the over extension. Upon review, we realize that the Behrenfeld et al. (2019) paper focuses on the global distribution and ecological aspects of diel vertical migration (DVM), but does not specifically address evolutionary patterns. We have revised the manuscript to reflect this more accurately and have removed the reference to evolutionary patterns in relation to their work.

"Furthermore, satellite-based lidar inversion of bbp signals has shown that zooplankton activity leads to pronounced bbp spikes, particularly at night, with these spikes most evident in the surface ocean layers, revealing the global distribution characteristics of diel vertical migration (DVM), as discussed in Behrenfeld et al. (2019)."

--Line 59-62 in highlight version

**Material and methods**

Line 72: I don't think it is correct to say in any way that pelagic fish are largely untapped resource for fisheries, especially in the North Atlantic! Maybe the authors refer to 'mesopelagic' here? I suggest clarifying or removing this bit of text.

AR: Thank you for your insightful comment. You are absolutely right that the reference to pelagic fish as an 'untapped resource' in the North Atlantic may not be accurate. We appreciate your suggestion to clarify this point. We have revised the text in the first paragraph of the Introduction to better reflect the importance of mesopelagic organisms and to correct any misleading statements about pelagic fish.

The revised text now emphasizes the significance of mesopelagic species in marine ecosystems and their potential contributions to fisheries resources, while avoiding any inaccurate claims about pelagic fish being largely untapped.

"The mesopelagic organisms, comprising species such as zooplankton, shrimp, squid, fish, and jellyfish, are estimated to harbor around one billion tonnes of biomass, representing a significant fraction of global fish biomass (Irigoien2014)"

--Line 22-23 in highlight version

Line 124: I suggest removing the use of the word 'pinnacle' and sticking with 'spikes'.

AR: Thank you for your suggestion. We agree that the term 'pinnacle' may be unclear in this context. We have revised the text to use 'spikes' instead, as it more accurately describes the observed patterns of aggregation for specific taxa.

"Frequency, defined as the number of spikes per unit depth, represents the likelihood of aggregation for specific taxa."

--Line 154-155 in highlight version

Figure 3 - I think this figure could be a great opportunity to provide a visual reference to the reader of how a profile of the metrics discussed in the results look like (e.g., frequency and density or spikes, etc.). I suggest adding them as a subfigure so that the reader has an immediate sense of how these metrics relate to the original 'spike' data. I also suggest using 2.0 as the maximum value of the bbp ratio as there are no points about 2.0 and going to 2.5 is a bit of a waste of precious space.

AR: Thank you for your suggestion. We have added a subfigure to Figure 3 to show how the metrics (e.g., spike frequency and density) relate to the original data. We also adjusted the bbp ratio maximum to 2.0. If there are any aspects that do not align with your suggestion or require further improvement, please do let us know.

Results - Lines 168-176: I think this content belongs to the discussion and not in the results.

AR: Thank you for your valuable suggestion. We will move this section to the Discussion part of the manuscript as per your recommendation.

Through the results: I think you are using here only daytime profiles but I'm not sure I could find easily this piece of information anywhere. If this is the case, please make sure it's explicit.

AR:Thank you for your comment. Both daytime and nighttime data were used in our study. To address your concern, we have explicitly stated this in the methods section and provided additional details to ensure clarity.

"To clarify the daily vertical migration of mesopelagic organisms, the water column was partitioned into 10-meter depth intervals, and profiles were categorized into daytime and nighttime based on the local solar time. In each interval, the spiking signals were normalized by calculating the proportion of spiking points relative to the total number of detected points, and the environmental factors were averaged."

--Line 146-149 in highlight version

Line 180: Why is there a reference to the Mediterranean Sea here?

AR:Thank you for your insightful comment. The reference to the Mediterranean Sea was removed because it lies outside our study area. Although we conducted experiments there, they fall beyond the scope of this paper. We appreciate your guidance and have made the necessary revisions.

Line following 185: Are the environmental variables used here from remote sensing or those measured from BGC floats?

AR: Thank you for your comment. The environmental variables used in this study include sea surface chlorophyll and sea surface temperature, which were derived from remote sensing data. The other environmental parameters, such as temperature, salinity, and oxygen levels, were obtained from BGC-Argo floats, which provide in situ measurements of these variables across the water column.

**Discussion**

Line ~205: When referring to the sources of variability of the position of DSL I recommend including the presence of mesoscale eddies as there is a good amount of work that showed they play quite a role in structuring DSL in the area: Fennel and Rose, 2015; Della Penna and Gaube, 2020; Devine et al., 2021.

AR: Thank you for your valuable suggestion. We fully acknowledge the significant role of mesoscale eddies in structuring the distribution of the deep scattering layer (DSL). Based on your recommendation, we have expanded the discussion to include the impact of eddies on DSL distribution and incorporated relevant references to enrich the manuscript. We have also conducted experiments regarding the influence of eddies. However, due to space and logical flow limitations, we could not detail these experiments in this paper. Nevertheless, we recognize the importance of this aspect and have supplemented the background information accordingly.

"Long-term factors such as food availability, light, oceanic physicochemical properties, and dissolved oxygen levels form the fundamental drivers (Fennell2015, Della2020, Devine2021). Short-term factors, including cloud cover, ocean currents, and lunar phases, also dynamically influence these behaviors (Lampert1989, Parra2019, Klevjer2020a, Hauss2016)."

--Line 70-72 in highlight version

Line 221: Why is there a reference to a 'mesocosm'? Please rephrase.

AR: Thank you for your guidance. We have made the correction and replaced "mesocosm" with "mesopelagic."

"The impact of these factors varies significantly across different regions and seasons (Klevjer2020a), leading to fluctuations in the mean intensity, intensity maxima distribution, and frequency of bbp spike layer signals within the mesopelagic layer."

--Line 263-264 in highlight version

Line 228: What is the 'mesopelagic acropora signal'?

AR:Thank you for pointing out the confusion regarding the term "mesopelagic acropora signal." We have revised the text to correct this and now refer to it as the "mesopelagic spike signal."

"In response, mesopelagic organisms migrate to the upper layers to exploit improved foraging opportunities, resulting in higher-frequency aggregations and a relative decrease in the mean intensity of the mesopelagic spike signal (Allan et al., 2021; Henson et al., 2012; Lutz et al., 2007; Woodd-Walker et al., 2002; Briggs et al., 2011; Vedenin et al., 2022)."

--Line 269-271 in highlight version

Line 272 and following: The impact of fronts on aggregations are not limited to downwelling and upwelling, so I suggest including here a mention as well of the horizontal mechanisms described in the next few lines. A very interesting discussion of light and fronts, water masses and zooplankton can also be found in Powell and Ohman, 2015.

AR: Thank you for your suggestion. We agree that the impact of fronts on mesopelagic aggregation includes not only vertical but also horizontal mechanisms, such as water mass interactions and mixing. We have revised the manuscript to include these horizontal processes, referencing relevant studies like Powell and Ohman (2015), to provide a more comprehensive understanding of how fronts, light, and mesopelagic organisms interact.

"Mesopelagic organisms also exhibit significant aggregation behaviors in frontal zones, where alternating downwelling and upwelling currents induce vertical displacements with substantial ecological impacts. In addition to these vertical mechanisms, mesoscale fronts also separate water masses through horizontal mixing, creating potential habitats for zooplankton (Martin, 2003). These horizontal processes, combined with light availability and nutrient dynamics, shape the spatial distribution of mesopelagic organisms and their aggregation behaviors in frontal zones (Powell and Ohman, 2015). Together, these mechanisms underscore the ecological complexity and significance of frontal regions."

--Line 369-374 in highlight version

**Reviewer: 2**

**Comments to the Author**

The manuscript presents an insightful exploration of the distribution characteristics and driving factors of mesopelagic organisms. However, there are several areas that require further refinement to enhance the clarity and credibility of the research.

AR: We truly appreciate your thorough review and valuable suggestions. The Materials and Methods section has been thoroughly revised to enhance clarity, with more detailed explanations and rationales provided for the methodologies employed.

**Major Concerns**

**I. Methods and Variable Specification**

The methods section falls short in providing adequate detail, hindering a thorough evaluation of the research approach and its reproducibility.

Comments 1: In lines 15 - 17, it is unclear whether the authors are referring to two distinct temperature - related variables: "temperature" and "vertical temperature gradient". Since the authors highlight temperature as the most impactful environmental factor, they should also provide the relative importance of these temperature - related variables. This information is crucial for understanding how each of these factors contributes to the distribution of mesopelagic organisms.

AR: Thank you for pointing this out. In the text, both sea surface temperature and vertical temperature gradient were mentioned. However, since the study focuses on mesopelagic organisms, which have large vertical migration ranges, the vertical temperature gradient has the greatest impact on them, with a relative importance of 26.03%. In contrast, the influence of sea surface temperature is minimal. My previous expression was inaccurate, and I have revised the original text accordingly.

"Random forest analysis revealed that the vertical temperature gradient was the most influential environmental factor affecting the distribution of mesopelagic organisms year - round, with a relative importance of 26.03%. Other critical factors include latitude, dissolved oxygen, salinity, mixed layer depth (MLD), and surface chlorophyll concentration, with relative importance values of 13.92%, 13.71%, 8.66%, 8.29%, and 8.09%, respectively."

**--Line 15-18 in highlight version**

Comments 2: In the study area description (lines 70 - 76), while the authors emphasize the North Atlantic's significance in carbon cycling and fishery resources, and touch on the potential implications of fish assemblage variations for ecological management, they fail to clearly link this background to the study area and the BGC - Argo - collected backscattering coefficient (bbp) profile data. To strengthen the study area selection rationale, quantitative analysis and literature support are needed. For instance, do mesopelagic organisms in this region exhibit unique or typical characteristics in carbon fixation efficiency, food web structure, or climate change response?

AR: Thank you very much for your insightful comments. We concur entirely that focusing solely on the importance of a specific fish species in the North Atlantic is both narrow and inaccurate. To provide a more comprehensive and accurate depiction, we have expanded our discussion to include the unique and typical characteristics of mesopelagic organisms in the North Atlantic, particularly in terms of carbon fixation efficiency, food web structure, and response to climate change. As demonstrated by Haěntjens et al. (2020), the backscattering coefficient (bbp) profiles collected by BGC-Argo floats are effective in reflecting the presence of mesopelagic zooplankton and sinking particle aggregates. Building on this foundation, we have incorporated additional literature to further enrich our analysis. The revised text is as follows:

"The North Atlantic (35° - 75°N, 0 - 70°W) is a critical region for global carbon cycling and marine ecosystems. Mesopelagic fish, through large-scale diel vertical migrations, efficiently transfer carbon and nutrients between the euphotic zone and deep sea, playing a key role in material cycling and energy flow (Lusher et al., 2016). Annually, demersal-pelagic fish on the UK-Ireland continental slope capture and store over one million tons of carbon dioxide equivalent (Trueman et al., 2014). Combined with planktonic regulation of carbon export via the biological carbon pump (Brun et al., 2019), these processes form the biological basis of the regional carbon cycle. The North Atlantic basin, spanning a large latitudinal gradient, experiences significant seasonal variations in solar radiation and primary production, which strongly influence mesozooplankton community dynamics. High-latitude overturning circulation and subduction processes of the Subtropical Circulation drive deep-sea dissolved organic carbon (DOC) transport, forming a key mechanism of the biological pump and contributing significantly to the global carbon cycle (Hansell et al., 2002; Falk-Petersen et al., 2009). Thus, the North Atlantic is central to addressing global climate change, preserving biodiversity, and guiding sustainable marine resource use. Figure 1 is based on backscattering coefficient (bbp) profiles collected by BGC-Argo floats across the study area, using  $1^{\circ} \times 1^{\circ}$  grid statistics."

--Line 82-93 in highlight version

Comments 3: According to research (e.g., Ljungström et al., 2021), light is a key factor influencing changes in mesopelagic organisms in mid - and high - latitude regions. However, the current model does not include this critical variable. Although the authors mention the impact of light on spatial distribution, the absence of light in the model prevents an objective and quantitative evaluation of its role. For instance, Ljungström et al. (2021) have shown that light significantly affects the vertical distribution and diel rhythms of mesopelagic organisms, with variations in light intensity and duration across different latitudes and seasons leading to substantial changes in their distribution. Therefore, the authors should seriously consider incorporating light as a variable into the model and presenting the corresponding quantitative results to provide a more comprehensive understanding of the distribution patterns of mesopelagic organisms.

AR: Thank you very much for your suggestions. The impact of light is indeed crucial for the distribution of mesopelagic organisms in mid- and high-latitude regions. We have incorporated PAR (photosynthetically active radiation) as a key variable into the random forest model and matched it with the results of mesopelagic organism aggregation. The results show that the variable importance of PAR is 8.66%, ranking only behind temperature and latitude. Additionally, we have supplemented and further explored the ways in which light affects these organisms in the discussion section, supported by several authoritative studies. We have endeavored to provide a comprehensive analysis of the potential drivers influencing the distribution of these organisms.

"For PAR, we utilized a high-resolution, long-term global gridded PAR product (2010–2018) provided by Tang (2021), which has a temporal resolution of three hours. Unlike solar altitude, which is based on latitude and time and may not fully capture the temporal and spatial variability in PAR, this dataset offers a more accurate and detailed representation of light availability."

--Line 109-112 in highlight version

"Random Forest variable importance analysis revealed that the vertical temperature gradient made the greatest contribution to the model, accounting for 26.03% of the variance. Following this, latitude (13.92%), dissolved oxygen at 500 m (13.71%), photosynthetically active radiation (PAR, 8.66%), salinity at 500 m (8.29%), mixed layer depth (MLD, 8.23%), average chlorophyll concentration (8.09%), average temperature (7.10%), and solar altitude (6.68%) were identified as the next most important factors. Among these, the vertical temperature gradient had the most significant impact on the seasonal and spatial distribution of mesopelagic organisms. Latitude, as a key geographical factor, also exerted a considerable influence on the spatial distribution patterns. Excluding the northeastern regions, mesopelagic organisms were generally found at shallower depths in higher latitudes. The model's response curves further elucidated the relationships between environmental factors and the aggregation depth of mesopelagic organisms in the open ocean. Within certain ranges, increasing latitude, higher dissolved oxygen levels, greater mixing, reduced light penetration, and decreasing temperatures all corresponded to shallower aggregation depths for midwater organisms. Across all regions, the distributions in summer and autumn tended to be shallower, whereas spring and winter distributions were generally deeper. These observations partially explain the consistency between the spatial distribution of midwater organisms and the heterogeneity of the physiological environment. In contrast, when considering the intensity of biological aggregation as a response variable, stronger signals from mesopelagic organisms typically originated from shallower depths. It is important to note that while Random Forest analysis can capture broad trends within specific ranges of environmental variability, the detailed seasonal differences across individual subregions require further multi-factorial analysis for a more comprehensive understanding."

"Other critical factors include latitude, dissolved oxygen, par, salinity, mld and surface chlorophyll concentration, with relative importance of 26.03%, 13.92%, 13.71%, 8.66%, 8.29%, 8.23% and 8.09%, respectively."

--Line 16-18 in highlight version

Comments 4: The North Atlantic's unique dynamic mixing processes, such as mixed layer depth and oceanic eddies, are not adequately addressed. The authors should clarify how they differentiate between the effects of these dynamic processes and other environmental factors on the aggregation of mesopelagic organisms.

AR: Following your recommendation, we have adopted the hybrid algorithm (Holte et al., 2017) to obtain more accurate estimates of the mixed layer depth (MLD). We have also conducted extensive work on the impacts of eddies and have achieved preliminary results. Since eddies primarily influence mesopelagic organisms indirectly through environmental changes driven by dynamic processes, the focus of this study is on the direct environmental factors affecting mesopelagic organisms. Therefore, we have not elaborated further on the eddy dynamics in this paper. However, in response to your suggestion, we have included the mixed layer in our analysis as a dynamic influencing factor. Additionally, we have expanded the discussion on the impact of fronts on mesopelagic organisms and added a description of eddy influence in the background introduction section of the introduction. The relevant description has been improved as follows.

"In addition to these key parameters, we incorporated two additional variables to enhance our analysis: Photosynthetically Active Radiation (PAR) and Mixed Layer Depth (MLD). These variables provide important insights into light conditions and the vertical structure of the ocean, both of which are critical for understanding the dynamics of mesopelagic organisms."

--Line 106-109 in highlight version

For MLD, we used data from the hybrid algorithm and threshold method (Holte et al., 2017). The hybrid algorithm was preferred for its accuracy, especially in regions like the Labrador and Irminger Seas, where the threshold method overestimates MLD by  $\sim 10\%$  in winter."

--Line 112-114 in highlight version

Despite significant advancements in understanding mesopelagic ecosystems, large-scale detection of mesopelagic organisms remains challenging, leading to considerable uncertainties in biomass estimates that range from billions to hundreds of tonnes (Gioesaeter et al., 1980; Sarant, 2014). The patterns of diel vertical migration (DVM) and seasonal vertical migration (SVM), their adaptive mechanisms, and the multifactorial influences on these behaviors are still poorly understood (Bandara et al., 2021). However, recent studies have shown that the aggregation and vertical migration of mesopelagic organisms are regulated by a complex interplay of multidimensional environmental variables. Long-term factors such as food availability, light, oceanic physicochemical properties, and dissolved oxygen levels form the fundamental drivers (Fennel and Rose, 2015; Della Penna and Gaube, 2020; Devine et al., 2021). Short-term factors, including cloud cover, ocean currents, and lunar phases, also dynamically influence these behaviors (Lampert et al., 1989; Parra et al., 2019; Klevjer et al., 2020a; Haass et al., 2016). Collectively, these findings indicate that the spatiotemporal distribution of mesopelagic organisms results from the interaction of macro-scale oceanic physical environments, micro-scale nutrient cycling, and periodic fluctuations. To address these challenges, we leveraged backscattering bbp and spike signals from BGC-Argo floats in the mid- and high-latitude regions of the North Atlantic. By examining diurnal and seasonal vertical migrations, analyzing horizontal distribution patterns, and identifying key environmental drivers using Random Forest modeling, we aimed to elucidate the mechanisms shaping mesopelagic ecosystems across diverse spatiotemporal scales.

--Line 65-79 in highlight version

Comments 5: It is unclear whether the methods employed in this study are based on existing algorithms with modifications. If improvements have been made, the details should be meticulously documented in the methods section. When compared to the study by Haëntjens et al. (2020), where only partial spike layers were presented, this study deals with a substantial number of profiles. It remains unclear whether all spike layers are combined in the analysis. If so, there may be an issue of interlacing layers at different depths, which could affect the accuracy of the averaged results. The authors need to further verify and explain this aspect.

AR: Thank you very much for your professional and targeted suggestions. We have indeed made some improvements to the algorithm used by the authors. It is important to clarify that we did not simply merge and average all the spike layers directly, as this would lead to misalignment between layers. For example, some layers might be located at 200 - 300 meters, while others are at 260 - 460 meters. Averaging all of them would mask local details, and the smoothed results would not show distinct spikes. Instead, we normalized the spike points of each spike layer across different depth ranges. For instance, if a spike layer contains three spike points, we recorded the position, time, and signal intensity of these points. The final statistical results

provide a more accurate representation of the aggregation situation. We have made more detailed modifications to the methods section to better explain this approach:

"Spike signals with identical features that occur simultaneously in two or more profiles are aggregated into a spike layer. To avoid misalignment of spike layer positions across different profiles during statistical analysis, we have made certain improvements to the reference method by extracting the internal spike point information of the aggregated layers. For each layer, we quantified the intensity, depth, and spike count of each spike point, which were then recorded for further analysis. The spike layer extraction workflow is illustrated in Figure 2."

--Line 124-130 in highlight version

**II. Results and Discussion**

Comments 6: In Figure 7, the y - axis lacks a clear definition, making it ambiguous to readers what variable or meaning it represents. Additionally, the latitude range from -100 to 100 appears questionable and requires re - examination to ensure data accuracy and appropriateness. To enhance the figure's comprehensibility and scientific rigor, the authors must provide a detailed explanation of the y - axis variable, including its specific meaning and unit, in the figure caption. This clarification is essential for readers to accurately interpret the results presented in the Random Forest model.

**AR**: Thank you for your comments. The y-axis in Figure 7 represents the anomaly in depth change of the primary spike layer. We have updated the figure caption accordingly.

"Response curves from the random forest model, with the blue line indicating the influence of various environmental factors. The small black ticks along the horizontal axis represent the distribution density of the data, while the gray points represent individual data points. The X-axis displays the range of feature values. The Y-axis shows the accumulated local effect (ALE) of each feature on the response variable (p), which reflects the anomaly in depth change of the primary spike layer. Positive values indicate a deepening of the spike, while negative values indicate a shoaling."

--Line 244 in highlight version

Comments 7 (Lines 186 - 200): The analysis of environmental driving factors, other than the temperature gradient, is rather cursory. The authors should expand on the underlying action mechanisms of these factors and their interactions with temperature across different seasons and regions.

**AR:** In response to your suggestion, we have expanded the analysis of environmental driving factors beyond the temperature gradient in both the results and discussion

sections. We have incorporated the roles of multiple environmental factors and have restructured the discussion to analyze their underlying mechanisms of action and interactions with temperature across different seasons and regions. Additionally, we have examined these factors from various perspectives, including diurnal, seasonal, and regional differences.

"Despite the limited availability of seasonal data, our observations across all regions reveal a consistent pattern: the vertical distribution of mesopelagic organisms is shallower during summer and autumn, and deeper during spring and winter. This trend is largely attributable to the light-driven seasonal patterns that govern mesopelagic organism distribution. The seasonal variations in the backscattering coefficient (bbp) spike layer intensity are influenced by a suite of environmental factors, including water temperature, ocean currents, dissolved oxygen levels, light availability, and food sources (Bianchi et al., 2013; Klevjer et al., 2016). The impact of these factors varies significantly across different regions and seasons (Klevjer et al., 2020a), leading to fluctuations in the mean intensity, intensity maxima distribution, and frequency of bbp spike layer signals within the mesopelagic layer. During spring and winter, the mean intensity of the bbp spike layer in the upper mesopelagic zone decreases, while its frequency increases relative to the middle mesopelagic layer. This shift is likely driven by the organisms' preference for specific depths influenced by lower temperatures, deeper mixed layers, limited light availability, and reduced phytoplankton concentrations in the upper layers during these seasons."

--Line 257-267 in highlight version

"As spring progresses and temperatures and light levels rise, the mixed layer becomes shallower and phytoplankton blooms increase. In response, mesopelagic organisms migrate to the upper layers to exploit improved foraging opportunities, resulting in higher-frequency aggregations and a relative decrease in the mean intensity of the mesopelagic spike signal (Allan et al., 2021; Henson et al., 2012; Lutz et al., 2007; Wood-Walker et al., 2002; Briggs et al., 2011; Vedenin et al., 2022). In the cooler months of spring and winter, strong downwelling increases surface water density, while salinity differences and stratification in high latitudes and the Atlantic Ocean facilitate the transfer of dissolved oxygen to deeper waters. Consequently, mesopelagic organisms migrate to greater depths in search of suitable habitats and food resources, thereby avoiding elevated predation pressure in surface waters (Freeman, 2006; Garcia-Soto et al., 2021; Yin et al., 2024). This migration results in a higher concentration of organisms in the middle layer and leads to a multilayer aggregation phenomenon. The correlation between dissolved oxygen in the 200–500 m layer and the negative correlation in the 500–800 m zone indicate a distinct oxygen minimum zone around 500-600 m, delineating the emergence of a prominent mesopelagic signal layer at approximately 600 m depth. During summer and autumn,

the mean frequency of bbp spike signals at depths shallower than 350 m is 1.85 and 4.15 times higher, respectively, than at greater depths. Notably, there is a pronounced aggregation of high-frequency signals in the near-surface layer, shallower than 50 m. In summer, a stable shallow mixed layer isolates the surface from deeper waters, concentrating mesopelagic organisms in the upper-middle layer. High-intensity and high-frequency signal layers emerge in the ocean's surface during summer and autumn. In autumn, these strong signals are frequently associated with chlorophyll maxima around 200 m depth. Increased solar radiation enhances phytoplankton photosynthesis, significantly boosting primary productivity and providing abundant food resources for larger marine organisms (Flombaum et al., 2013). Warmer sea surface temperatures also create favorable conditions for species thriving in warmer waters, promoting the survival, reproduction, and growth of larger marine organisms (Chen et al., 2019; Bova et al., 2021). Additionally, ocean circulation and upwelling transport nutrient-rich deep waters to the surface, attracting larger marine species to feed during the day."

--Line 2657-288 in highlight version

"Our analysis of bbp spike signal frequency and intensity reveals significant seasonal differences between the upper and middle layers of the ocean. In spring and winter, although the average bbp spike intensity in the upper ocean is lower than in the middle layer (where peak values are primarily distributed), mesopelagic organisms still aggregate at specific depths in the middle layer and migrate to the upper ocean for foraging. In contrast, in summer and autumn, especially summer, both the average intensity and frequency of bbp spikes are significantly higher in the upper layer than in the middle layer, with a marked concentration in the near-surface zone. This shift indicates a seasonal change in mesopelagic behavior, with a heightened preference for upper-layer habitats and foraging during warmer months. A similar pattern in the mesopelagic scatterers of intermediate to deep layers was noted by Powell and Ohman (2015), who investigated the scattering characteristics of migratory and non-migratory zooplankton in frontal regions. Their study found that shallower migratory layers, which consist of smaller but more abundant scatterers, are more homogeneously distributed at finer scales. In contrast, deeper non-migratory layers likely consist of fewer but larger scatterers, and these are associated with a lower abundance of organisms, which are likely non-migratory in nature. The 400-500 m depth range of the mesopelagic layer, typically inhabited by non-swimming species or crustaceans, is shaped by vertical fluxes of organic carbon and particulate matter (Marohn et al., 2021; Liu, 2011; Sikder et al., 2019; Henson et al., 2012; Lutz et al., 2007). Based on our findings, lower intensity but higher frequency signals may correspond to smaller-sized plankton or particle-based signals, while higher intensity and lower frequency signals are likely associated with larger, but fewer, organisms. This distribution pattern may be driven by multiple mechanisms: First, larger mesopelagic organisms, with stronger swimming abilities, tend to migrate to deeper waters to avoid currents, while smaller organisms remain in the upper layers (Lin and Costello, 2023; Sorochan et al., 2023). Second, during spring and winter, the

deeper mixed layer and unstable water column in the North Atlantic, along with transient stratification events often disrupted by storms, favor the accumulation of organic matter in the deeper mixed layer, resulting in increased biotic aggregation frequencies in the mid-ocean (Dall'Olmo et al., 2016). These mechanisms collectively shape the vertical distribution and seasonal dynamics of mesopelagic organisms, providing new insights into the structure and function of marine ecosystems."

--Line 289-310 in highlight version

Comments 8 (Lines 235 - 240): The concept of small high - frequency signals corresponding to smaller organisms and large low - frequency signals associated with larger organisms is introduced in the discussion section but not pre - introduced in the methods or data section. Visually marking these portions in the relevant figures would enhance the interpretability of the results.

**AR**: Thank you for your suggestion. We have added a subfigure to Figure 3 to illustrate the relationship between the metrics (e.g., spike frequency and density) and the original data. Correspondingly, we have revised lines 181 - 182 to provide a detailed explanation of the correspondence between the signals in the figure and the original data. The figure has also been updated to enhance the readability of the results.

Figure 3. This figure illustrates the diurnal distribution of bbp signals and environmental factors, with colored lines indicating daytime and grey lines representing nighttime. Specifically, figure a depicts bbp signals, figure b shows chlorophyll levels.

Comments 9: The discussion on seasonal vertical migration lacks connections with

other crucial processes in the marine ecosystem during the same period. The authors should delve into the relationships between seasonal migration patterns and the overall functioning of the marine ecosystem and supplement their findings with additional supporting results.

**AR:** Thank you for your valuable feedback. We have revised the discussion on seasonal vertical migration to better connect it with other crucial processes in the marine ecosystem. Specifically, we have added the following passage to highlight the interrelationships:

"As spring progresses and temperatures and light levels rise, the mixed layer becomes shallower and phytoplankton blooms increase. In response, mesopelagic organisms migrate to the upper layers to exploit improved foraging opportunities, resulting in higher-frequency aggregations and a relative decrease in the mean intensity of the mesopelagic spike signal (Allan et al., 2021; Henson et al., 2012; Lutz et al., 2007; Wood-Walker et al., 2002; Briggs et al., 2011; Vedenin et al., 2022)."

--Line 268-271 in highlight version

**Minor Points**

Comments 10: Throughout the text, it is recommended to insert a space between the main text and references to enhance readability.

AR: Thank you for pointing this out. We have reviewed the entire manuscript and have inserted spaces between the main text and references wherever necessary to enhance readability. The corrections have been made consistently throughout the document.

Line 96: "The SST data,"

Line 117: "Previous studies (Haëntjens et al., 2020) to show that..."

Line 206: "Previous studies (Loisel et al., 2002) for indicating that..."

Line 345: "Braun et al., 2023) In high-latitude regions..."

Line 354: Previous studies have shown that (Chawarski et al., 2022).

Line 356: "similar to previous studies (Norheim et al., 2016). (Langbehn et al., 2022) found that in high latitudes..." "Previous studies (Kaartvedt, 2008) indicate that..."

Comments 11: Some of the terms utilized in the manuscript may not be familiar to all

readers. For example, "spiking layer signals" mentioned in the abstract (lines 5 - 6) should be substituted with a more commonly used term, such as "bbp spikes", to improve the text's clarity and accessibility.

AR: Thank you for your suggestion. We agree that using more widely recognized terms can enhance the clarity and accessibility of our manuscript. We have replaced the term "spiking layer signals" with "bbp spikes" in the abstract (lines 5 - 6) and have reviewed the rest of the manuscript to ensure that all terminology is clear and familiar to a broader audience.

Comments 12: In lines 35 - 37, the authors mention traditional methods for studying mesopelagic organisms, such as trawl and acoustic methods. However, it is unclear whether these methods are used for detection or sampling. The authors should provide a clear and explicit statement about the specific use of these methods to avoid confusion. For example, they could state, "Traditional methods for detecting and sampling mesopelagic organisms, including trawl sampling and acoustic surveys..."

AR: Thank you for your helpful comment. We have revised the text in lines 35 - 37 to provide a clearer and more explicit statement regarding the use of these methods. The revised sentence now reads: "Traditional methods for detecting and sampling mesopelagic organisms, including trawl sampling and acoustic surveys, have been widely used in previous studies..." This clarification aims to avoid any confusion regarding the specific applications of these methods.

In response to the reviewer's suggestion, we have conducted a thorough review of the manuscript and made additional revisions to enhance the content and logical flow. Specifically, we have refined the logical content of the introduction and supplemented the methodology section as follows:

"In recent years, significant progress has been made in utilizing backscattering coefficient (bbp) spike signals from BGC-Argo floats to study marine biological processes. These signals have shown a strong correlation with mesopelagic biological information, as evidenced by their high concordance with acoustic trawl observations (Haentjens et al., 2020). Specifically, the bbp spike signals are mainly produced by larger particles that are closely related to biological aggregations (Briggs et al., 2011). For instance, the extensive diatom blooms in the North Atlantic each spring lead to a substantial increase in particulate matter, consisting of fresh phytoplankton aggregates that rapidly sink to the seafloor (Lampitt, 1985; Honjo and Manganini, 1993). Occasionally, large spikes in optical profiles are also interpreted as aggregates or zooplankton (Bishop et al., 1999; Gardner et al., 2000; Bishop and Wood, 2008). The bbp signal captures the entire particle assemblage, including zooplankton, detritus, bacteria, and mineral particles. Notably, significant increases in bbp are observed when small zooplankton dominate the mixed layer community (Rembauville et al., 2017; Petit, 2023). Furthermore, satellite-based lidar inversion of

bbp signals has revealed that zooplankton activity can cause pronounced bbp spikes, particularly at night, with these spikes being most evident in the surface ocean layers. This finding sheds light on the global distribution characteristics of diel vertical migration (DVM) of zooplankton (Behrenfeld et al., 2019). Collectively, these studies indicate that  $b_6$  spike signals not only reflect the presence of large particulate matter and tiny zooplankton but also capture the diel vertical migration of zooplankton, providing a powerful tool for understanding marine biological dynamics."

--Line 50-64 in highlight version

"Furthermore, the bbp spike signals we analyzed include not only zooplankton but also spikes from sinking material aggregates with high precision (>90%) (Haëntjens et al., 2020). The spike layer extraction workflow is illustrated in Figure 2."

--Line 128-130 in highlight version

**References (reintegrated in the paper)**

Aksnes, D. L., Dupont, N., Staby, A., Fiksen, Ø., Kaartvedt, S., & Aure, J. (2009). Coastal water darkening and implications for mesopelagic regime shifts in Norwegian fjords. Marine Ecology Progress Series, 387, 39-49.

Aksnes, D. L., Røstad, A., Kaartvedt, S., Martinez, U., Duarte, C. M., & Irigoien, X. (2017). Light penetration structures the deep acoustic scattering layers in the global ocean. Science advances, 3(5), e1602468.

Berge, J., Varpe, Ø., Moline, M. A., Wold, A., Renaud, P. E., Daase, M., & Falk-Petersen, S. (2012). Retention of ice-associated amphipods: possible consequences for an ice-free Arctic Ocean. Biology Letters, 8(6), 1012-1015.

Braun, C. D., Della Penna, A., Arostegui, M. C., Afonso, P., Berumen, M. L., Block, B. A., ... & Thorrold, S. R. (2023). Linking vertical movements of large pelagic predators with distribution patterns of biomass in the open ocean. Proceedings of the National Academy of Sciences, 120(47), e2306357120.

Brun, P., Stamieszkin, K., Visser, A. W., Licandro, P., & Payne, M. R. (2019). Climate change has altered zooplankton-fuelled carbon export in the North Atlantic. Nature Ecology & Evolution, 3(3), 416–423. https://doi.org/10.1038/s41559-018-0780-3

Catalá, T. S., Reche, I., Álvarez, M., Khatiwala, S., Guallart, E. F., Benítez-Barrios, V. M., ... & Álvarez-Salgado, X. A. (2015). Water mass age and aging driving chromophoric dissolved organic matter in the dark global ocean. Global Biogeochemical Cycles, 29(7), 917-934.

Chawarski, J., Klevjer, T. A., Cote, D., & Geoffroy, M. (2022). Evidence of temperature control on mesopelagic fish and zooplankton communities at high latitudes. Frontiers in Marine Science, 9, 917985.

Della Penna, A., & Gaube, P. (2020). Mesoscale eddies structure mesopelagic communities. Frontiers in Marine Science, 7, 454.

Devine, B., Fennell, S., Themelis, D., & Fisher, J. A. (2021). Influence of anticyclonic, warm-core eddies on mesopelagic fish assemblages in the Northwest Atlantic Ocean. Deep Sea Research Part I: Oceanographic Research Papers, 173, 103555.

Falk-Petersen, S., Mayzaud, P., Kattner, G., & Sargent, J. R. (2009). Lipids and life strategy of Arctic Calanus. Marine Biology Research, 5(1), 18-39. https://doi.org/10.1080/17451000802512267

Fennell, S., & Rose, G. (2015). Oceanographic influences on deep scattering layers across the North Atlantic. Deep Sea Research Part I: Oceanographic Research Papers, 105, 132-141.

Gillooly, J. F., Brown, J. H., West, G. B., Savage, V. M., & Charnov, E. L. (2001). Effects of size and temperature on metabolic rate. science, 293(5538), 2248-2251.

Grimaldo, E., Grimsmo, L., Alvarez, P., Herrmann, B., Møen Tveit, G., Tiller, R., ... & Selnes, M. (2020). Investigating the potential for a commercial fishery in the Northeast Atlantic utilizing mesopelagic species. ICES Journal of Marine Science, 77(7-8), 2541-2556.

Gu, P., Liu, Z., & Delworth, T. L. (2024). Strong oceanic forcing on decadal surface temperature variability over global ocean. Geophysical Research Letters, 51(8), e2023GL107401.

Gu, S., Liu, Z., Ng, H. C., Lynch-Stieglitz, J., McManus, J. F., Spall, M., ... & Wu, L. (2024). Open ocean convection drives enhanced eastern pathway of the Glacial Atlantic Meridional Overturning Circulation. Proceedings of the National Academy of Sciences, 121(45), e2405051121.

Holte, J., L. D. Talley, J. Gilson, and D. Roemmich (2017), An Argo mixed layer climatology and database, Geophys. Res. Lett., 44, 5618–5626, doi:10.1002/2017GL073426.

Ingvaldsen, R. B., Eriksen, E., Gjøsæter, H., Engås, A., Schuppe, B. K., Assmann, K. M., ... & Bluhm, B. A. (2023). Under-ice observations by trawls and multi-frequency acoustics in the Central Arctic Ocean reveals abundance and composition of pelagic fauna. Scientific Reports, 13(1), 1000.

Kaartvedt, S. (2008). Photoperiod may constrain the effect of global warming in arctic marine systems. Journal of plankton research, 30(11), 1203-1206.

Klevjer, T. A., Irigoien, X., Røstad, A., Fraile-Nuez, E., Benítez-Barrios, V. M., & Kaartvedt, S. (2016). Large scale patterns in vertical distribution and behaviour of mesopelagic scattering layers. Scientific reports, 6(1), 19873.

Klevjer, T. A., Melle, W., Knutsen, T., & Aksnes, D. L. (2020). Vertical distribution and migration of mesopelagic scatterers in four north Atlantic basins. Deep Sea Research Part II: Topical Studies in Oceanography, 180, 104811.

Klevjer, T., Norheim, E., Aksnes, D., Strand, E., Knutsen, T., Melle, W., & Wiebe, P. (2015). Zooplankton and micronekton vertical distribution and diel vertical migration behaviour in the Northern North Atlantic Ocean.

Langbehn, T. J., Aksnes, D. L., Kaartvedt, S., Fiksen, Ø., Ljungström, G., & Jørgensen, C. (2022). Poleward distribution of mesopelagic fishes is constrained by seasonality in light. Global Ecology and Biogeography, 31(3), 546-561.

Luo, J., Ortner, P. B., Forcucci, D., & Cummings, S. R. (2000). Diel vertical migration of zooplankton and mesopelagic fish in the Arabian Sea. Deep Sea Research Part II: Topical Studies in Oceanography, 47(7-8), 1451-1473.

Lusher, A. L., O'Donnell, C., Officer, R., & O'Connor, I. (2016). Microplastic interactions with North Atlantic mesopelagic fish. ICES Journal of marine science, 73(4), 1214-1225. https://doi.org/10.1093/ICESJMS/FSV241

Martin, A. P. (2003). Phytoplankton patchiness: the role of lateral stirring and mixing. Progress in oceanography, 57(2), 125-174.

Nelson, N. B., & Siegel, D. A. (2013). The global distribution and dynamics of chromophoric dissolved organic matter. Annual review of marine science, 5(1), 447-476.

Norheim, E., Klevjer, T. A., & Aksnes, D. L. (2016). Evidence for light-controlled migration amplitude of a sound scattering layer in the Norwegian Sea. Marine Ecology Progress Series, 551, 45-52.

Powell, C. F., Baker, A. R., Jickells, T. D., Bange, H. W., Chance, R. J., & Yodle, C. (2015). Estimation of the atmospheric flux of nutrients and trace metals to the eastern tropical North Atlantic Ocean. Journal of the Atmospheric Sciences, 72(10), 4029-4045.

Powell, J. R., & Ohman, M. D. (2015). Changes in zooplankton habitat, behavior, and acoustic scattering characteristics across glider-resolved fronts in the Southern California Current System. Progress in Oceanography, 134, 77-92.

Proud, R., Cox, M. J., & Brierley, A. S. (2017). Biogeography of the global ocean's mesopelagic zone. Current Biology, 27(1), 113-119.

Puerta, P., Johnson, C., Carreiro-Silva, M., Henry, L. A., Kenchington, E., Morato, T., ... & Orejas, C. (2020). Influence of water masses on the biodiversity and biogeography of deep-sea benthic ecosystems in the North Atlantic. Frontiers in Marine Science, 7, 239.

Robinson, C., et al. (2010). Mesopelagic zone ecology and biogeochemistry – a synthesis. Deep-Sea Research Part II: Topical Studies in Oceanography, 57(15), 1504-1518.

Sallée, J. B., Pellichero, V., Akhoudas, C., Pauthenet, E., Vignes, L., Schmidtko, S., ... & Kuusela, M. (2021). Summertime increases in upper-ocean stratification and mixed-layer depth. Nature, 591(7851), 592-598.

Saupe, E. E., Myers, C. E., Townsend Peterson, A., Soberón, J., Singarayer, J., Valdes, P., & Qiao, H. (2019). Spatio-temporal climate change contributes to latitudinal diversity gradients. Nature ecology & evolution, 3(10), 1419-1429.

Smetacek, V., & Nicol, S. (2005). Polar ocean ecosystems in a changing world. Nature, 437(7057), 362-368.

Tang, W. (2021). A long-term and high-resolution global gridded photosynthetically active radiation product (1984-2018). National Tibetan Plateau / Third Pole Environment Data Center. https://doi.org/10.11888/RemoteSen.tpdc.271909. https://cstr.cn/18406.11.RemoteSen.tpd c.271909.

Tang, W., Qin, J., Yang, K., Jiang, Y., & Pan, W. (2022). Mapping long-term and high-resolution global gridded photosynthetically active radiation using the ISCCP H-series cloud product and reanalysis data. Earth System Science Data, 14(4), 2007-2019.

Trueman, C. N., Johnston, G., O'hea, B., & MacKenzie, K. M. (2014). Trophic interactions of fish communities at midwater depths enhance long-term carbon storage and benthic production on continental slopes. Proceedings of the Royal Society B: Biological Sciences, 281(1787), 20140669. https://doi.org/10.1098/rspb.2014.0669

---

## Author Response (AR2)

**Author's Response:**

In response to the reviewers' second-round comments, we have carefully verified the variable importance values presented in the abstract. We identified inaccuracies in the relative importance percentages for salinity and mixed layer depth (MLD). Accordingly, we have corrected these values to 8.29% and 8.23% respectively (Line 18). Additionally, we have incorporated the importance value for photosynthetically active radiation (PAR) into the text.

The modifications are as follows:

**Original version:**

"Random forest analysis revealed that the vertical temperature gradient was the most influential environmental factor affecting the distribution of mesopelagic organisms year-round, with a relative importance of 26.03%. Other critical factors include latitude, dissolved oxygen, salinity, mixed layer depth (MLD), and surface chlorophyll concentration, with relative importance values of 13.92%, 13.71%, 8.66%, 8.29%, and 8.09%, respectively."

**Revised version:**

"Random forest analysis revealed that the vertical temperature gradient was the most influential environmental factor affecting the distribution of mesopelagic organisms year-round, with a relative importance of 26.03%. Other critical factors include latitude, dissolved oxygen, photosynthetically active radiation, salinity, mixed layer depth (MLD), and surface chlorophyll concentration, with relative importance values of 13.92%, 13.71%, 8.66%, 8.29%, 8.23%, and 8.09%, respectively."